# DLGNet: Hyperedge Classification via a Directed Line Graph for Chemical Reactions

## Abstract

Graphs and hypergraphs provide powerful abstractions for modeling interactions among a set of entities of interest and have been attracting a growing interest in the literature thanks to many successful applications in several fields, including chemistry. In the paper, we address the reaction classification task by introducing the *Directed Line Graph* (DLG) transformation for directed hypergraphs. Building on this representation, we propose the Directed Line Graph Network (DLGNet), the first spectral-based Graph Neural Network (GNN) designed to perform convolutions directly on the DLG. At the core of DLGNet lies a novel complex-valued Hermitian matrix, the *Directed Line Graph Laplacian* ($\vec{\mathbb{L}}_N$), which effectively encodes directional interactions within the hypergraph's structure through the DLG representation. Experimental results on three real-world chemical reaction datasets demonstrate that DLGNet consistently outperforms all baseline competitors.

## 1 Introduction

Graph representations have recently been applied in chemistry and biology to address various tasks such as drug discovery (Bongini et al., 2021), molecule generation (Hoogeboom et al., 2022), and protein interaction modeling (Jha et al., 2022). Focusing on chemical reactions, several graph-based representations have been developed and employed with applications in areas such as reaction engineering, retrosynthetic pathway design, and reaction feasibility evaluations. Retrosynthetic modeling, in particular, where a synthetic route is designed starting from the desired product and analyzed backward, benefits greatly from accurate reaction type identification. This capability enables the elimination of unfeasible pathways, thereby streamlining the discovery of efficient routes for chemical production. This is particularly important in industries such as pharmaceutical and material sciences, where optimizing synthetic routes can lead to significant cost savings and enable innovation. A similar situation holds in reaction feasibility analysis, where predicting the likelihood of a reaction's success based on molecular inputs is essential for designing scalable and efficient processes.

One of the most relevant techniques to model reactions relies on a *directed graph* (Fialkowski et al., 2005) where molecules are represented as nodes and the chemical reactions are represented as directed edges from reactants to products. Despite its popularity, such a model suffers from a key limitation, since modeling each reaction as a collection of *individual* directed edges between each reactant-product pair fails to fully capture the complexity of multi-reactant or multi-product reactions, which are crucial in many applications (Restrepo, 2022; Garcia-Chung et al., 2023). To mitigate this issue, Restrepo (2023); Chang (2024) introduced a *directed hypergraph* representation which is able to model both the chemical reactions structure and their directionality, where directed hyperedges model the directional interactions between reagents (heads) and products (tails), better capturing the full complexity of chemical reactions, see Figure 1 left. Indeed, hypergraphs generalize the notion of a graph by allowing *hyper*edges to connect an arbitrary number of nodes, thereby capturing both pairwise (dyadic) and group-wise (polyadic) interactions (Schaub et al., 2021).

In contrast to prior studies that address node classification or link prediction tasks (Dong et al., 2020; Wang et al., 2023b; Zhao et al., 2024), in this work we tackle the reaction classification task—i.e., predicting the reaction type of a given set of reactants and products—as a *hyperedge classification* task. A naive approach for a graph model would involve combining a pair of node feature vectors and passing them to a classifier. However, in hypergraphs, this is not straightforward since each

$e1 : b + c \longrightarrow a \quad e2 : a + b \longrightarrow d \quad e3 : e \longrightarrow d$

Figure 1: Three reactions and their corresponding directed hypergraph representation (left). The hypergraph is then transformed into its directed line graph (right). The hyperedges of $\vec{H}$ become the nodes of $\mathrm{DLG}(\vec{H})$ and are connected if they overlap in $\vec{H}$. Complex-valued edge weights in $\mathrm{DLG}(\vec{H})$ encode $\vec{H}$'s directionality, as detailed in Section 3.

hyperedge contains a varying number of nodes, making direct feature concatenation impractical. To address this, *undirected* hyperedge classification can be reformulated as a node classification problem by constructing a *line graph*, a graph in which each hyperedge in the original hypergraph is transformed into a node. In this way, classification can be performed on the newly defined nodes, overcoming the limitation of pairing feature vectors, effectively solving the hyperedge classification problem.

In the paper, we address the classification problem for chemical reactions following the recent literature where such reactions are modeled via directed hyperedges. For this purpose, we formally introduce the concept of the *Directed Line Graph* of a given *weighted directed* hypergraph $\vec{H}$, denoted as *DLG($\vec{H}$)*. In $\mathrm{DLG}(\vec{H})$, the vertices correspond to the hyperedges of $\vec{H}$, and a directed edge connects two vertices whenever the corresponding hyperedges in $\vec{H}$ share at least one vertex, as shown in Figure 1. Since the nodes of $\mathrm{DLG}(\vec{H})$ correspond to hyperedges of $\vec{H}$, this modeling approach allows us to directly manipulate hyperedge features, which are critical for solving the reaction-classification task. In order to carry out such manipulation with the framework of convolutional neural networks of the spectral type, we define the *Directed Line Graph Laplacian* $\vec{\mathbb{L}}_N$, a Laplacian matrix which is specifically designed to capture both directed and undirected adjacency relationships between the hyperedges of $\vec{H}$ via its directed line graph $\mathrm{DLG}(\vec{H})$. We prove that $\vec{\mathbb{L}}_N$ enjoys different key properties, among which being Hermitian (i.e., being a complex-valued matrix with a symmetric real part and a skew-symmetric imaginary one) and being positive semidefinite. These properties allow us to introduce a spectral convolutional operator for $\mathrm{DLG}(\vec{H})$. We rely on $\mathrm{DLG}(\vec{H})$ and its Laplacian matrix as the foundations of the Directed Line Graph Network (DLGNet), the first (to our knowledge) spectral-based GNN designed for the convolution of hyperedge features rather than node features.

For the task of *hyperedge classification in the chemical reaction domain*, our experimental results show that transitioning from directed hypergraphs to directed line graphs and performing convolutions directly on the latter is beneficial. DLGNet achieves an average relative percentage difference improvement of 2.91% over the second-best method across a collection of real-world datasets, with a maximum improvement of 4.21%. We also carry out an extensive set of ablation studies, which confirm the importance of the various components of DLGNet.

**Main Contributions of This Work**

- We address the *hyperedge classification task in the chemical reaction domain* by introducing the formal definition of a directed line graph associated with a weighted directed hypergraph $\vec{H}$, the *Directed Line Graph* $\mathrm{DLG}(\vec{H})$, on which we rely to capture a set of higher-order relationships between a set of molecules. We then propose the Directed Line Graph Laplacian $\vec{\mathbb{L}}_N$, a Hermitian matrix that captures both directed and undirected relationships between the hyperedges of a weighted directed hypergraph via its DLG. We also prove that $\vec{\mathbb{L}}_{\mathbb{N}}$ enjoys many desirable spectral properties.

- To tackle the hyperedge classification task, we introduce DLGNet, the first spectral-based GNN specifically designed to operate on directed line graphs associated with weighted directed hypergraphs by directly convolving hyperedge features rather than node features.

- We perform an extensive experimental evaluation on the reaction classification task on three real-world datasets. Our results highlight the advantages of our approach compared to other methods presented in the literature.

## 2 GRAPH AND HYPERGRAPH LEARNING BACKGROUND

An undirected hypergraph is defined as an ordered pair $H = (V, E)$, with $n := |V|$ and $m := |E|$, where $V$ is the set of vertices (or nodes) and $E \subseteq 2^V \setminus \{\}$ is the (nonempty) set of hyperedges. The weights of the hyperedges are stored in the diagonal matrix $W \in \mathbb{R}^{m \times m}$, where $w_e$ is the weight of hyperedge $e \in E$ (in the unweighted case we have $W = I$). The vertex degree $d_u$ and hyperedge degree $\delta_e$ are defined as $d_u := \sum_{e \in E: u \in e} |w_e|$ for $u \in V$, and $\delta_e := |e|$ for $e \in E$. These degrees are stored in two diagonal matrices $D_v \in \mathbb{R}^{n \times n}$ and $D_e \in \mathbb{R}^{m \times m}$. In the case of 2-uniform hypergraphs (i.e., graphs), the matrix $A \in \mathbb{R}^{n \times n}$ is defined such that $A_{uv} = w_e$ for each $e = \{u, v\} \in E$ and $A_{uv} = 0$ otherwise; we refer to it as the *adjacency* matrix of the graph. Hypergraphs where $\delta(e) = k$ for some $k \in \mathbb{N}$ for all $e \in E$ are called $k$-uniform. Following Gallo et al. (1993), we define a directed hypergraph $\vec{H}$ as a hypergraph where each node in each hyperedge $e \in E$ belongs to either a *head set* $H(e)$ or a *tail set* $T(e)$. If $T(e)$ is empty, $e$ is an undirected hyperedge.

The relationship between vertices and hyperedges in a undirected hypergraph $H$ is classically represented via an incidence matrix $B$ of size $n \times m$, where $B$ is defined as:

$$B_{ve} = \begin{cases} 1 & \text{if } v \in e \\ 0 & \text{otherwise} \end{cases} \qquad v \in V, e \in E. \tag{1}$$

From the incidence matrix $B$, one can derive the *Signless Laplacian Matrix $Q$* as well as its normalized version $Q_N$ (Chung &Graham, 1997):

$$Q := BWB^\top \qquad Q_N := D_v^{-\frac{1}{2}} BW D_e^{-1} B^\top D_v^{-\frac{1}{2}}, \tag{2}$$

where $W, D_e, D_v$ are the diagonal matrices defined above. Following Zhou et al. (2006), the Laplacian for a general undirected hypergraph is defined as:

$$\Delta := I - Q_N. \tag{3}$$

The Laplacian matrix encodes the hypergraph's connectivity and hyperedge weights.

Letting $\mathcal{L}$ be a generic Laplacian matrix of a given 2-uniform hypergraph $H$ and following Kipf &Welling (2017), we rely on the following convolution operator:

$$\hat{Y} := \theta_0 I + \theta_1 \mathcal{L}. \tag{4}$$

For a detailed description of how this operator is derived and its relationship to graph Fourier transforms, see Appendix B

## 3 THE DIRECTED LINE GRAPH AND ITS LAPLACIAN

As stated in the introduction, in this paper we tackle the task of molecule-reaction classification as a hyperedge classification problem by re-framing it as a node classification task, leveraging the concept of the *line graph*. The *line graph* $L(H)$ of an undirected hypergraph $H$ is classically defined as the undirected graph whose vertex set is the hyperedge set of $H$. In $L(H)$, two vertices $i, j$ are adjacent—i.e., $L(H)$ contains the edge $\{i, j\}$—if and only if their corresponding hyperedges $i, j$ have a nonempty intersection (Tyshkevich &Zverovich, 1998). By construction, $L(H)$ is a 2-uniform graph. Its adjacency matrix is defined as:

$$A(L(H)) := \mathbb{Q} - W D_e, \tag{5}$$

where $\mathbb{Q} := B^\top B$ is, by construction, the Signless Laplacian of $L(H)$. [1] The normalized version of $\mathbb{Q}$ and the corresponding normalized Laplacian are defined as:

$$\mathbb{Q}_N := D_e^{-\frac{1}{2}} W^{\frac{1}{2}} B^\top D_v^{-1} B W^{\frac{1}{2}} D_e^{-\frac{1}{2}}$$
$$\mathbb{L}_N := I - \mathbb{Q}_N. \tag{6}$$

---

[1]This is because the incidence matrix of $L(H)$ is $B^*$.

Notice that, from equation 2, one can define the weighted version of $B$ as $BW^{\frac{1}{2}}$. The definitions in equation 6 rely on the same matrix, but transposed.

To the best of our knowledge, the literature does not offer any formal definition for the *line graph* associated with a weighted directed hypergraph $\vec{H}^2$. To address this, we first define a complex-valued incidence matrix $\vec{B}$ which preserves the inherent directionality of $\vec{H}$:

$$\vec{B}_{ve} := \begin{cases} 1 & \text{if } v \in H(e), \\ -\mathrm{i} & \text{if } v \in T(e), \qquad v \in V, e \in E. \\ 0 & \text{otherwise.} \end{cases} \tag{7}$$

The idea is to generalize the classical construction of $\mathbb{Q} = B^T B$ of the Signless Laplacian of the line graph of a 2-uniform graph to the case of a hypergraph by leveraging our proposed incidence matrix $\vec{B}$. First, we propose the following definition:

**Definition 1.** The Directed Line Graph $DLG(\vec{H})$ of a weighted directed hypergraph $\vec{H}$ is a 2-uniform hypergraph whose vertex set corresponds to the hyperedge set of $\vec{H}$ and whose adjacency matrix is the following complex-valued skew-symmetric matrix:[3]

$$A(DLG(\vec{H})) = W^{\frac{1}{2}} \vec{B}^* \vec{B} W^{\frac{1}{2}} - W D_e. \tag{8}$$

Using equation 8 of definition 1 and Equations equation 5–equation 6, we obtain the following formulas for the normalized Signless Laplacian $\vec{\mathbb{Q}}_N \in \mathbb{C}^{m \times m}$ and the normalized Laplacian $\vec{\mathbb{L}}_N \in \mathbb{C}^{m \times m}$ of $DLG$, which we refer to as *Signless Directed Line-Graph* and *Directed Line Graph Laplacian*:

$$\vec{\mathbb{Q}}_N := \vec{D}_e^{-\frac{1}{2}} W^{\frac{1}{2}} \vec{B}^* \vec{D}_v^{-1} \vec{B} W^{\frac{1}{2}} \vec{D}_e^{-\frac{1}{2}}$$
$$\vec{\mathbb{L}}_N := I - \vec{\mathbb{Q}}_N. \tag{9}$$

To the best of our knowledge, the *Directed Line Graph Laplacian* has not been explored in the existing literature.

To better understand how $\vec{\mathbb{L}}_N$ encodes the directionality of $\vec{H}$, we illustrate its definition in scalar form for a pair of hyperedges $i, j \in E$ (which correspond to vertices in $DLG(\vec{H})$):

$$\vec{\mathbb{L}}_N(ij) = \begin{cases} 1 - \sum_{u \in i} \dfrac{w_i}{d_u \delta_i} & i = j \\ \left( -\sum_{\substack{u \in H(i) \cap H(j) \\ \vee u \in T(i) \cap T(j)}} \dfrac{w_i^{\frac{1}{2}} w_j^{\frac{1}{2}}}{d_u} - \mathrm{i} \sum_{u \in H(i) \cap T(j)} \dfrac{w_i^{\frac{1}{2}} w_j^{\frac{1}{2}}}{d_u} + \mathrm{i} \sum_{u \in T(i) \cap H(j)} \dfrac{w_i^{\frac{1}{2}} w_j^{\frac{1}{2}}}{d_u} \right) \dfrac{1}{\delta_i^{\frac{1}{2}}} \dfrac{1}{\delta_j^{\frac{1}{2}}} & i \neq j \end{cases} \tag{10}$$

When $i = j$, we are in the self-loop part of the equation and $\vec{\mathbb{L}}_N(ij)$ weights hyperedge $i$ proportionally to its weight $w_i$ and inversely proportionally to its density and the density of its nodes. When $i \neq j$, $\vec{\mathbb{L}}_N(ij)$'s value depends on the interactions between the hyperedges of $\vec{H}$ (which correspond to the nodes of $DLG(\vec{H})$). Let $u \in V$ be a node and $i, j \in E$ be two hyperedges in the hypergraph $\vec{H}$. If $u$ belongs to the head set of both hyperedges (i.e., $u \in H(i) \cap H(j)$) or to the tail set of both (i.e., $u \in T(i) \cap T(j)$), its contribution to the real part of $\mathbb{L}_N(ij)$, $\Re(\vec{\mathbb{L}}_N(ij))$, is negative. For the undirected line graph associated with an undirected hypergraph, this is the only contribution, consistent with the behavior of $\mathbb{L}_N$ (as described in equation 6). If $u$ takes opposite

---

[2]The line graph introduced in Bretto (2013) is defined exclusively for unweighted directed hypergraphs. Moreover, edges in that line graph are established only between hyperedges (i.e., nodes of the line graph) that share vertices under specific head–tail configurations. As a consequence, the construction neglects head–head and tail–tail relationships, which are instead encompassed by our definition.

[3]Notice that this matrix underlies a 2-uniform graph with complex-valued edge weights; to our knowledge, the literature offers no Laplacian matrix suitable for such a case besides the Laplacian operator we propose in this paper.

roles in hyperedges $i$ and $j$, i.e, it belongs to the head set in $i$ and to the tail set in $j$ or *vice versa*, it contributes to the imaginary part of $\mathbb{L}_N$, $\Im(\vec{\mathbb{L}}_N(ij))$, negatively when $u \in H(i) \cap T(j)$, and positively when $u \in T(i) \cap H(j)$. Consequently, $\Im(\vec{\mathbb{L}}_N(ij))$ coincides with the *net* contribution of all the vertices that An example illustrating the construction of $\vec{\mathbb{L}}_N$ for a directed line graph associated with a weighted directed hypergraph is provided in Appendix I. Let us point out that the behavior of *Directed Line Graph Laplacian* does not coincide (to the best of our knowledge) with any of the Laplacian matrices previously proposed in literature (see Appendix C for more details).

With the following theorem, we show that $\vec{\mathbb{L}}_N$ is a generalization of $\mathbb{L}_N$ (defined in equation 6) from the undirected to the directed case:

**Theorem 1.** *If $\vec{H}$ is undirected (i.e., $\vec{H} = H$), $\vec{\mathbb{L}}_N = \mathbb{L}_N$ and $\vec{\mathbb{Q}}_N = \mathbb{Q}_N$ holds.*

The *Directed Line Graph Laplacian* enjoys several properties. First, to be able to adopt our Laplacian within a convolution operator in line with Kipf &Welling (2017) and other literature approaches Zhang et al. (2021); Fiorini et al. (2023), we must show that our Laplacian is positive semidefinite. For this, we derive the expression of the 2-Dirichlet energy function associated with it. Such a function coincides with the Euclidean norm $||x||^2_{\vec{\mathbb{L}}_N}$ induced by $\vec{\mathbb{L}}_N$ for a signal $x \in \mathbb{C}^m$:

**Theorem 2.** *Letting $\mathbf{1}$ be the indicator function, the Euclidean norm induced by $\vec{\mathbb{L}}_N$ of a complex-valued signal $x = a + ib \in \mathbb{C}^m$ with a component per hyperedge in $E$ reads:*

$$\frac{1}{2} \sum_{u \in V} \frac{1}{d_u} \sum_{i,j \in E} \left[ \left( (\psi a_i - \phi a_j)^2 + (\psi b_i - \phi b_j)^2 \right) \mathbf{1}'(u) \right.$$
$$\left. + \left( (\psi a_i - \phi b_j)^2 + (\phi a_j + \psi b_i)^2 \right) \mathbf{1}''(u) + \left( (\psi a_i + \phi b_j)^2 + (\phi a_j - \psi b_i)^2 \right) \mathbf{1}'''(u) \right], \quad (11)$$

where $\psi = \frac{w_j}{\delta_i}^{\frac{1}{2}}, \phi = \frac{w_i}{\delta_j}^{\frac{1}{2}}$, $\mathbf{1}'(u) = \mathbf{1}_{u \in H(i) \cap H(j) \vee u \in T(i) \cap T(j)}$, $\mathbf{1}''(u) = \mathbf{1}_{u \in H(i) \cap T(j)}$, and $\mathbf{1}'''(u) = \mathbf{1}_{u \in T(i) \cap H(j)}$. Since the function in Theorem 2 is a real-valued sum of squares, we deduce the following spectral property for $\vec{\mathbb{L}}_N$:

**Corollary 3.** *$\vec{\mathbb{L}}_N$ is positive semidefinite.*

Next, we show next that $\vec{\mathbb{Q}}_N$ has a nonnegative spectrum:

**Theorem 4.** *$\vec{\mathbb{Q}}_N$ is positive semidefinite.*

By applying Corollary 3 and Theorem 4, we derive upper bounds on the spectra of $\vec{\mathbb{L}}_N$ and $\vec{\mathbb{Q}}_N$:

**Corollary 5.** *$\lambda_{\max}(\vec{\mathbb{L}}_N) \leq 1$ and $\lambda_{\max}(\vec{\mathbb{Q}}_N) \leq 1$.*

The proofs of the theorems and corollaries of this section can be found in Appendix C.

## 4 DIRECTED LINE GRAPH NETWORK (DLGNET)

The properties of the proposed Laplacian make it possible to derive a well-defined spectral convolution operator from it. In this work, this operator is integrated into the Directed Line Graph Network (DLGNet). Specifically, based on equation 4, by setting $\mathcal{L} = \vec{\mathbb{L}}_N$, the convolution operator is defined as $\hat{Y}x = \theta_0 I + \theta_1 \vec{\mathbb{L}}_N$. The advantage of adopting two parameters $\theta_0, \theta_1$ within DLGNet's localized filter is explained by the following result:

**Proposition 6.** *The convolution operator derived from equation 4 by setting $\mathcal{L} = \vec{\mathbb{L}}_N$ with parameters $\theta_0$ and $\theta_1$ is the same as the convolution operator obtained by using $\mathcal{L} = \vec{\mathbb{Q}}_N$ with parameters redefined as $\theta_0' = \theta_0 + \theta_1$ and $\theta_1' = -\theta_1$.*

This shows that, by selecting appropriate values for $\theta_0$ and $\theta_1$, DLGNet can leverage either $\vec{\mathbb{L}}_N$ or $\vec{\mathbb{Q}}_N$ as convolution operator to maximize the performance on the task at hand.

We define $X \in \mathbb{C}^{m \times c_0}$ as a $c_0$-dimensional graph signal (a graph signal with $c_0$ input channels), which we compactly represent as a matrix. This matrix serves as the feature matrix of the hyperedges of $\vec{H}$ which we construct from the feature matrix of the nodes $X' \in \mathbb{C}^{n \times c_0}$ of $\vec{H}$. Specifically, inspired by the operation used in the *reduction component* for graph pooling (Grattarola et al., 2022), we define the feature matrix for the vertices of $DLG(\vec{H})$ as $X = \vec{B}^* X'$. This approach combines features through summation, based on the topology defined by $\vec{B}$. See Appendix F for more details.

In our network, the scalar parameters $\theta_0$ and $\theta_1$ are subsumed by two operators $\Theta_0, \Theta_1 \in \mathbb{C}^{c_0 \times c}$ which we use to carry out a linear transformation on the feature matrix $X$. A similar transformation, which can also increase or decrease the number of channels of $X$, is adopted in other GNNs such as MagNet (Zhang et al., 2021). DLGNet features $\ell$ convolutional layers. The output $Z \in \mathbb{C}^{m \times c'}$ of any such layer adheres to the following equation:

$$Z(X) = \phi\left(IX\Theta_0 + \vec{\mathbb{L}}_N X\Theta_1\right), \tag{12}$$

where $\phi$ is the activation function. Following Fiorini et al. (2023; 2024), DLGNet employs a complex-valued *ReLU* where $\phi(z) = z$ if $\Re(z) \geq 0$ and $\phi(z) = 0$ otherwise, with $z \in \mathbb{C}$. DLGNet also utilizes a residual connection for every convolutional layer except the first one, a choice which helps prevent oversmoothing and has been proven to be helpful in a number of works, including He et al. (2016); Kipf & Welling (2017). After the convolutional layers, following Zhang et al. (2021), we apply an *unwind* operation where we transform $Z(X) \in \mathbb{C}^{m \times c'}$ into $(\Re(Z(X)) || \Im(Z(X))) \in \mathbb{R}^{m \times 2c'}$, where $||$ is the concatenation operator. To obtain the final results, DLGNet features $S$ linear layers, with the last one employing a Softmax activation function.

For more information on the inference complexity of DLGNet and its expressive power, see Appendix D.

## 5 EXPERIMENTAL RESULTS

In this section, we present three real-world datasets, the baseline models, and the results on the reaction classification task.

**Datasets.** We test DLGNet on common organic chemistry reaction classes, namely various chemical transformations that are fundamental to both research and industrial chemistry. Those include molecular rearrangements, such as the interconversion (substitution) or the elimination of molecular substituents, as well as the introduction of specific functional groups (e.g., acyl, alkyl, or aryl groups) in a chemical compound. Other important reaction classes involve the formation of certain bond-types (e.g., carbon-carbon: C–C) or structures (e.g., heterocyclic compounds). We rely on a standard dataset (`Dataset-1`) and additionally construct two new ones (`Dataset-2` and `Dataset-3`)—see Figure 3 in Appendix E. Further details about the datasets can be found in Appendix E.

*Dataset-1.* As the main source of data, we use the reactions from USPTO granted patents (Lugo-Martinez et al., 2021), which is the most widely used dataset for retrosynthesis problems and contains about 480K reactions. After removing duplicates and erroneous reactions, we select a subset, namely `Dataset-1`, comprising 50K atom-mapped reactions belonging to 10 different classes. An example component from `Dataset-1` is reported in Figure 3 (Appendix E), left upper panel.

*Dataset-2.* This dataset is the result of the merging of data from five different sources and contains 5300 reactions. It presents a smaller number of reaction types, but a larger variety of substituents and reaction conditions, such as the presence of solvent or catalyst, hence providing additional complexity on some specific classes for the model to predict. Figure 3, upper right panel, illustrates an example from it. Given that some elements are shared across the data sources, we combine them into three major classes.

*Dataset-3.* Since the two datasets listed so far only include single-product reactions, in order to test the model on a highly complex task we add a third collection, `Dataset-3`, comprised of double-product bimolecular nucleophilic substitution ($S_N2$) and triple-product bimolecular elimination (E2) reaction classes, extracted from von Rudorff et al. (2020) and totaling 649 competitive reactions. A schematic representation of `Dataset-3`'s elements is reported in Figure 3, lower panel.

**Features.** Each node in the hypergraph corresponds to a molecule. To effectively represent molecular characteristics, we assigned specific features to each node. Across the three datasets, we employ the Morgan Fingerprints (MFs) Rogers &Hahn (2010) as features. MFs are among the most widely used molecular descriptors, designed to encode a molecule by identifying the presence or absence of specific substructures (fragments) within its molecular graph. The algorithm iteratively updates the representation of each atom based on its local chemical environment, considering neighboring atoms within a predefined radius. A radius of $r$ indicates that structural information up to $r$ bonds away from each atom is incorporated into the final fingerprint representation.

**Baselines.** We employ baselines from two main groups: HNNs that only handle undirected hypergraphs and HNNs which are specifically designed for directed hypergraphs. Similar to GNNs, HNNs are either spatial- or spectral-based. Spatial-based HNNs treat the convolution operator as a localized aggregation function (Dong et al., 2020). On the other hand, spectral-based HNNs define the convolution operator (based on graph signal processing and graph Fourier transforms) as a function of the eigenvalue decomposition of the Laplacian matrix associated with the hypergraph Feng et al. (2019).

In the spectral-based category, methods such as HGNN (Feng et al., 2019), HCHA in Dong et al. (2020), and HGNN$^+$ (Gao et al., 2022) are analogous to GNNs applied to clique expansions of hypergraphs. For spatial-based methods, HNNs such as HNHN (Dong et al., 2020), UniGCNII (Huang &Yang, 2021), HyperDN (Tudisco et al., 2021), LEGCN Yang et al. (2022), as well as set-based models (Chien et al., 2021), AllDeepSets and AllSetTransformer, incorporate hyperedge features and employ a message-passing framework, and can be interpreted as GNNs applied to the star expansion graph. Additionally, ED-HNN (Wang et al., 2023a) leverages gradient diffusion processes to generalize across a broad class of hypergraph neural networks, while PhenomNN (Wang et al., 2023b) introduces a framework based on hypergraph-regularized energy functions.

In the context of directed HNNs, we consider two state-of-the-art models: DHM (Zhao et al., 2024) and DHRL (Ma et al., 2024). The first one, DHM, encodes high-order information in directed hypergraphs and captures the directional information of directed hyperedges through an attention mechanism and a directed hypergraph momentum encoder. The second method, DHRL, approximates the Laplacian of the directed hypergraph and formulates the convolution operation on this directed hypergraph structure. Differently from these baselines, the proposed DLGNet is a spectral-based GCN that leverages the Directed Line Graph Laplacian and convolves on the directed line graph derived from the directed hypergraph (see Section 3). Further details on the computational are reported in the Appendix F.

### 5.1 EXPERIMENTAL DETAILS

The 13 state-of-the-art baseline methods we consider have been adapted to address the hyperedge classification task. This adaptation is necessary to adjust the output dimensionality—specifically, through a pooling operation—to match that of the hyperedges. This adaptation follows a standardized mechanism across all methods, which is derived from DLGNet. Specifically, it involves inserting the linear feature transfer operator, $X = \vec{B}^* X'$, as described in Section 3 (with further details provided in Appendix F). This operator is applied downstream of the convolution block and is followed by $\ell$ linear layers.

The hyperparameters of these baselines and of our proposed model are selected via grid search (see Appendix F). The datasets are split into 50% for training, 25% for validation, and 25% for testing. The experiments are conducted with 5 random data splits and the average F1-score along with the standard deviation across the splits is reported. We choose the F1-score as evaluation metric due to the class imbalance naturally present in the datasets. Throughout the tables contained in this section, the best results are reported in **boldface** and the second best are underlined. The datasets and code we used are available on GitHub (see Appendix A).

### 5.2 RESULTS

**Quantitative.** The F1-score along with the relative standard deviation across different methods, datasets, and folds is presented in Table 1. The results show that, across the three datasets, DLGNet achieves an average additive performance improvement over the best-performing competitor of

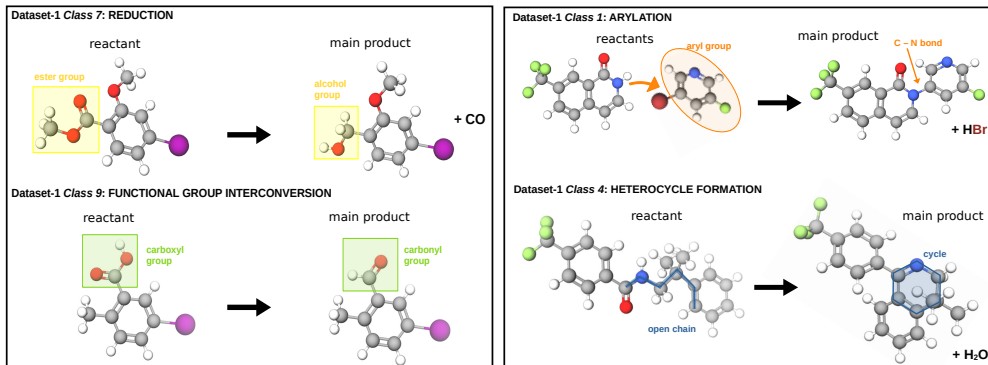

Figure 2: Ball-and-stick 3D model of `Dataset-1` mislabeled pairs of reaction classes. Color code: grey for carbon, red for oxygen, blue for nitrogen, purple for iodine, green for chlorine, light green for fluorine, brown for bromide, and white for hydrogen. (**Left panel, upper**): Reduction from a ester to an alcohol substituent on a 6-carbon atoms ring. (**Left panel, lower**): Functional group interconversion from carboxyl to carbonyl group in the analog hexagonal structure. (**Right panel, upper**): arylation reaction between an amine compound and a aryl halide, yielding a C–N bond in the final product. (**Right panel, lower**): heterocycle formation via amide intramolecular condensation, producing a hexagonal ring containing a heteroatom (nitrogen).

approximately 2.95 percentage points. In terms of Relative Percentage Difference (RPD)[4], we have an average RPD improvement of 2.91%. DLGNet achieves the largest performance gain on `Dataset-2`, with a relative percentage difference (RPD) improvement of approximately 4.21% and an average absolute improvement of 4.31 percentage points over the second-best competitor. HNN-based methods for undirected hypergraphs underperform on two out of three datasets compared to approaches specifically designed for directed hypergraphs, such as DHM and DHRL. Notably, our proposed DLGNet, which applies convolutions directly on the DLG, achieves superior performance, outperforming all competitors.

Table 1: Mean F1-score and standard deviation obtained on the hyperedge classification task.

| Topology | Method | Dataset-1 | Dataset-2 | Dataset-3 |
|---|---|---|---|---|
| | HGNN | $21.86 \pm 1.52$ | $79.89 \pm 3.97$ | $55.72 \pm 5.89$ |
| | HCHA/HGNN$^+$ | $21.26 \pm 0.69$ | $80.22 \pm 3.91$ | $64.70 \pm 4.16$ |
| | HCHA w/ Attention | $21.86 \pm 1.52$ | $34.91 \pm 1.21$ | $39.33 \pm 3.69$ |
| | HNHN | $14.35 \pm 0.24$ | $60.95 \pm 14.8$ | $41.49 \pm 4.36$ |
| Hypergraph | UniGCNII | $14.33 \pm 0.26$ | $72.58 \pm 0.96$ | $41.64 \pm 4.39$ |
| | HyperND | $13.96 \pm 0.29$ | $75.69 \pm 0.52$ | $36.66 \pm 5.40$ |
| | AllDeepSets | $14.37 \pm 0.24$ | $72.15 \pm 1.02$ | $37.79 \pm 6.55$ |
| | AllSetTransformer | $14.37 \pm 0.24$ | $72.58 \pm 1.06$ | $37.62 \pm 6.50$ |
| | ED-HNN | $14.35 \pm 0.27$ | $72.98 \pm 1.81$ | $37.82 \pm 6.49$ |
| | PhenomNN | $14.34 \pm 0.28$ | $75.01 \pm 1.23$ | $40.11 \pm 4.73$ |
| Directed Hypergraph | DHM | $46.04 \pm 0.58$ | $59.31 \pm 4.04$ | $68.10 \pm 3.60$ |
| | DHRL | $58.15 \pm 1.58$ | $79.36 \pm 3.94$ | $99.27 \pm 0.79$ |
| Directed-Line Graph | **DLGNet** | $\mathbf{60.55 \pm 0.80}$ | $\mathbf{83.67 \pm 3.41}$ | $\mathbf{99.75 \pm 0.34}$ |

**Qualitative.** To gain deeper insights into the capability of DLGNet of classifying different reaction types, we analyze the confusion matrices for `Dataset-1` and `Dataset-2`. The results of this analysis are presented in Figure 4 and Figure 5 in Appendix H. The confusion matrix for `Dataset-1` reveals that, while most of the classes are predicted with high accuracy, e.g., Protection and Functional group addition reactions (accuracy of 88% and 77%, respectively), some are predicted not as well, e.g., Functional group interconversion (41%). To better understand this behavior, we conducted a thorough inspection of the structural features of `Dataset-1`'s components, selecting several

---
[4]The RPD of two values $P_1, P_2$ is the percentage ratio of their difference to their average, i.e., $|P_1 - P_2|/\frac{P_1+P_2}{2}\%$.

Table 2: Ablation study. Average F1-score and standard deviation are reported.

| Method | Dataset-1 | Dataset-2 | Dataset-3 |
|---|---|---|---|
| **DLGNet** | **60.55 ± 0.80** | **83.67 ± 3.41** | **99.75 ± 0.34** |
| DLGNet w/o dir | 52.07 ± 1.61 | 70.19 ± 0.65 | 81.65 ± 8.39 |
| DLGNet w/ Signless Laplacian | 60.24 ± 0.36 | 82.86 ± 1.96 | **99.75 ± 0.55** |
| DLGNet w/ $\Theta_0 = 0$ | 53.82 ± 0.74 | 75.68 ± 3.59 | 91.45 ± 2.36 |
| DLGNet w/o skip-conn | 56.38 ± 3.02 | 80.63 ± 3.54 | 99.63 ± 0.34 |

elements from pairs of classes among which the model yields the highest uncertainty. Two example cases are reported in Figure 2. Overall, our analysis reveals that the pair of classes that are subject to the higher degree of confusion are, structurally, highly similar, which well explains the poorer performance that DLGNet achieves on them, as we illustrate in the following. The left panel illustrates the mislabeling of *Class 9* (Functional group interconversions, correctly predicted in 41% of the cases) with *Class 7* (Reductions, incorrectly predicted in 14% of the cases), while the right panel presents an example of *Class 4* (Heterocycle formations, correctly predicted in 44% of the cases) with *Class 1* (Arylations, incorrectly predicted 30% of the cases). Notably, in these examples, both the main backbone structure of the molecules and the substituent groups (the segments affected by the reactive process, highlighted in the figure) exhibit a high degree of similarity between the two classes. In the left panel, the reactants of both classes present a 6-carbon ring (in grey) as well as a iodine substituent (in purple). The atoms composing the highlighted groups are also of the same types. On the other hand, in the right panel, the majority of the constituent parts of the products are in common between the two classes. Specifically, despite the outcome of *Class 4* is the formation of a heterocycle, i.e., a hexagonal ring containing a heteroatom (nitrogen, in blue), such a geometrical feature is also present in *Class 1* arylation product, as the resulting molecule presents two heterocycles rings. Similar considerations apply to the incorrect labeling of Dataset-2 N-arylation sub classes, where the main difference between the reactants lies in the nature of the aryl halide that participates in the coupling reaction. Overall, our model demonstrates strong predictive performance across the majority of the classes, although a few, particularly those with shared elements, remain challenging to differentiate. Nevertheless, we are confident that DLGNet will prove highly valuable to the chemistry community, allowing for the categorization of existing data sources as well as for planning new synthetic routes.

**Ablation study.** Table 2 presents the results of an ablation study carried out on DLGNet to assess the importance of directionality in DLGNet's line graph. To do this, we test DLGNet using an undirected line graph (DLGNet w/o dir) and demonstrate that DLGNet consistently outperforms it on all three data sets. This indicates that directionality is crucial in solving the chemical reaction classification task. Focusing on equation 12, we test DLGNet under two conditions: *i)* using $\vec{\mathbb{Q}}_N$ instead of $\vec{\mathbb{L}}_N$ (DLGNet w/ Signless Laplacian), and *ii)* setting $\Theta_0 = 0$ (DLGNet w/ $\Theta_0 = 0$), thus nullifying the first term in equation 12. The first comparison shows identical results across all datasets, thus providing a computational confirmation of the results of Proposition 6, while DLGNet w/ $\Theta_0 = 0$ performs worse. Finally, we assess the architectural choice related to the incorporation of skip connections. While DLGNet without skip connections (DLGNet w/out skip-conn) exhibits a slight drop in performance, the results remain close to those of the original architecture.

## 6 CONCLUSIONS

In this paper, we tackled the molecular reaction classification problem as a hyperedge classification problem by introducing the Directed Line Graph Network (DLGNet), the first spectral GNN specifically designed to operate on directed line graphs associated with directed hypergraphs. DLGNet leverages a novel complex-valued Laplacian matrix, the *Directed Line Graph Laplacian*, which is a Hermitian matrix encoding the interactions among the hyperedges of a hypergraph using complex numbers. This formulation enables the natural representation of both directed and undirected relationships between the hyperedges, capturing rich structural information. Through extensive evaluation on the chemical reaction classification problem using three real-world datasets, we have demonstrated the consistent superiority of DLGNet. Through an ablation study, we demonstrated the relevance of encoding directional information via the directed line graph.

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

## A  CODE REPOSITORY AND LICENSING

The code developed for this research work is available at `https://anonymous.4open.science/r/HyperedgeClassification-2E63` and freely distributed under the Apache 2.0 license.[5] code for the baselines used in the experimental analysis is available at `https://github.com/Graph-COM/ED-HNN`, `https://github.com/yxzwang/PhenomNN` and `https://github.com/WBZhao98/DHMConv` under the MIT license.[6]

## B  DERIVATION OF THE CONVOLUTION OPERATOR *a là* KIPF&WELLING

We assume that $\mathcal{L}$ admits an eigenvalue decomposition $\mathcal{L} = U\Lambda U^*$, where $U \in \mathbb{C}^{n \times n}$ represents (in its columns) the eigenvectors, $U^*$ is its conjugate transpose, and $\Lambda \in \mathbb{R}^{n \times n}$ is the diagonal matrix containing the eigenvalues. Let $x \in \mathbb{C}^n$ be a *graph signal*, i.e., a complex-valued function $x : V \to \mathbb{C}^n$ of the vertices of $H$. We define $\hat{x} = \mathcal{F}(x) = U^*x$ as the *graph Fourier transform* of $x$ and $\mathcal{F}^{-1}(\hat{x}) = U\hat{x}$ its inverse transform. The convolution $y \circledast x$ between $x$ and another graph signal $y \in \mathbb{C}^n$, acting as a *filter*, in the vertex space is defined in the frequency space as $y \circledast x = U\text{diag}(U^*y)U^*x$. Letting $\hat{Y} := U\hat{G}U^*$ with $\hat{G} := \text{diag}(U^*y)$, we can write $y \circledast x$ as the linear operator $\hat{Y}x$. See Shuman et al. (2013) for more details.

In the context of GNNs, explicitly learning $y$ as a *non-parametric filter* presents two significant limitations. Firstly, computing the eigenvalue decomposition of $\mathcal{L}$ can be computationally too expensive (Kipf &Welling, 2017). Secondly, explicitly learning $y$ requires a number of parameters proportional to the input size, which becomes inefficient for high-dimensional tasks (Defferrard et al., 2016). To address these issues, the GNN literature commonly employs filters where the graph Fourier transform is approximated as a degree-$K$ polynomial of $\Lambda$, with a small $K$ for computational efficiency. For further details, we refer the reader to Kipf &Welling (2017); Defferrard et al. (2016); Huang et al. (2024).

This way, we obtain a so-called *localized filter*, thanks to which the output (i.e., filtered) signal at a vertex $u \in V$ is a linear combination of the input signals within $K$ edges of $u$ (Shuman et al., 2013). By employing various polynomial filters and setting $K = 1$ (as virtually in the the whole of the GCN literature Kipf &Welling (2017); Zhang et al. (2021); Fiorini et al. (2023)), such as Chebyshev polynomials as in Hammond et al. (2011); Kipf &Welling (2017) or power monomials as used

---

[5] `https://www.apache.org/licenses/LICENSE-2.0`

[6] `https://choosealicense.com/licenses/mit/`

by Singh &Chen (2022), one obtains a parametric family of linear operators with two learnable parameters, $\theta_0$ and $\theta_1$: [7]

$$\hat{Y} := \theta_0 I + \theta_1 \mathcal{L}.$$

## C  PROPERTIES OF OUR PROPOSED LAPLACIAN

This section contains the proofs of the theorems and corollaries reported in the main paper.

**Theorem 1.** *If $\vec{H}$ is undirected (i.e., $\vec{H} = H$), $\vec{\mathbb{L}}_N = \mathbb{L}_N$ and $\vec{\mathbb{Q}}_N = \mathbb{Q}_N$ holds.*

*Proof.* Since $H = (V, E)$ is an undirected hypergraph, $\vec{B}$ is binary and only takes values 0 and 1 (rather than being ternary and taking values $0, 1, -i$), defining an undirected line graph $L(H)$. In particular, for each edge $e \in E$ we have $\vec{B}_{ue} = 1$ if either $u \in H(e)$ or $u \in T(e)$ and $\vec{B}_{ue} = 0$ otherwise. Consequently, the directed incident matrix $\vec{B}$ is identical to the non-directed incidence matrix $B$, i.e., $\vec{B} = B$. Thus, by construction, $\vec{\mathbb{L}}_N = \mathbb{L}_N$ and $\vec{\mathbb{Q}}_N = \mathbb{Q}_N$. $\square$

**Theorem 2.** *Letting $\mathbf{1}$ be the indicator function, the Euclidean norm induced by $\vec{\mathbb{L}}_N$ of a complex-valued signal $x = a + ib \in \mathbb{C}^m$ with a component per hyperedge in $E$ reads:*

$$\frac{1}{2} \sum_{u \in V} \frac{1}{d_u} \sum_{i,j \in E} \left[ \left( (\psi a_i - \phi a_j)^2 + (\psi b_i - \phi b_j)^2 \right) \mathbf{1}'(u) \right.$$
$$\left. + \left( (\psi a_i - \phi b_j)^2 + (\phi a_j + \psi b_i)^2 \right) \mathbf{1}''(u) + \left( (\psi a_i + \phi b_j)^2 + (\phi a_j - \psi b_i)^2 \right) \mathbf{1}'''(u) \right], \quad (13)$$

*Proof.* By definition, we have

$$x^* \vec{\mathbb{L}}_N x = x^* I x - x^* \vec{\mathbb{Q}}_N x = x^* I x - x^* \vec{D}_e^{-\frac{1}{2}} W^{\frac{1}{2}} \vec{B}^* \vec{D}_v^{-1} \vec{B} W^{\frac{1}{2}} \vec{D}_e^{-\frac{1}{2}}.$$

Scalarly, the expression reads

$$\sum_{i \in E} x_i^* x_i - \sum_{i,j \in E} \sum_{u \in V} \frac{1}{d_u} \frac{w_i^{\frac{1}{2}} \vec{B}_{ui}^* \vec{B}_{uj} w_j^{\frac{1}{2}}}{\delta_i^{\frac{1}{2}} \delta_j^{\frac{1}{2}}} x_i^* x_j,$$

where $\sum_{i,j \in E}$ indicates the sum over all ordered pairs $i, j$ in $E$, including those where $i = j$. W.l.o.g., we can swap the order of the sums in the second term, obtaining:

$$\sum_{i \in E} x_i^* x_i - \sum_{u \in V} \sum_{i,j \in E} \frac{1}{d_u} \frac{w_i^{\frac{1}{2}} \vec{B}_{ui}^* \vec{B}_{uj} w_j^{\frac{1}{2}}}{\delta_i^{\frac{1}{2}} \delta_j^{\frac{1}{2}}} x_i^* x_j.$$

Due to $\mathbb{Q}_N$ being Hermitian, $\mathbb{Q}_N + \mathbb{Q}_N^* = 2\mathbb{Q}_N$ holds. Thus, substituting $\frac{1}{2}(\mathbb{Q}_N + \mathbb{Q}_N^*)$ for $\mathbb{Q}_N$, we can rewrite the second term as

$$-\frac{1}{2} \sum_{u \in V} \frac{1}{d_u} \sum_{i,j \in E} w_i^{\frac{1}{2}} \left( \vec{B}_{ui}^* \vec{B}_{uj} \frac{x_i^* x_j}{\delta_i^{\frac{1}{2}} \delta_j^{\frac{1}{2}}} + \vec{B}_{uj}^* \vec{B}_{ui} \frac{x_j^* x_i}{\delta_j^{\frac{1}{2}} \delta_i^{\frac{1}{2}}} \right) w_j^{\frac{1}{2}}.$$

Next, we show that the following holds for the first term:

$$\sum_{i \in E} x_i^* x_i = \sum_{u \in V} \frac{1}{d_u} \sum_{i,j \in E: u \in i \wedge u \in j} w_j \frac{x_i^* x_i}{\delta_i}.$$

---

[7] Following w.l.o.g. Singh &Chen (2022), we employ the approximation $\hat{G} = \sum_{k=0}^{K} \theta_k \Lambda^k$, from which we deduce $Yx = U\hat{G}U^*x = U(\sum_{k=0}^{K} \theta_k \Lambda^k)U^*x = \sum_{k=0}^{K} \theta_k (U\Lambda^k U^*)x = \sum_{k=0}^{K} \theta_k \mathcal{L}^k x.$

We show this by showing how to turn the right-hand side into the left-hand side. First, we pre-pone the sum over $i$ in the right-hand side, obtaining:

$$\sum_{i \in E} \left( \sum_{u \in V} \frac{1}{d_u} \sum_{j \in E: u \in j} w_j \frac{x_i^* x_i}{\delta_i} \right) .$$

Then, we bring $\frac{1}{\delta_i}$ and $x_i^* x_i$ outside of the inner summation, which leads to the following expression

$$= \sum_{i \in E} x_i^* x_i \frac{1}{\delta_i} \underbrace{\sum_{u \in V} \frac{1}{d_u} \underbrace{\sum_{j \in E: u \in j} w_j}_{=1}}_{=1} .$$

Following the calculations reported as underbraces, we deduce that the coefficient that multiplies $x_i^* x_j$ is equal to 1, concluding this part of the proof.

As we did for the second term, we now double the summation in the first term and compensate for it with a factor of $\frac{1}{2}$, obtaining:

$$\frac{1}{2} \sum_{u \in V} \frac{1}{d_u} \sum_{i,j \in E: u \in i \wedge u \in j} \left( w_j \frac{x_i^* x_i}{\delta_i} + w_i \frac{x_j^* x_j}{\delta_j} \right) .$$

Looking back at both terms, we have the expression:

$$\frac{1}{2} \sum_{u \in V} \frac{1}{d_u} \sum_{i,j \in E: u \in i \wedge u \in j} \left( w_j \frac{x_i^* x_i}{\delta_i} + w_i \frac{x_j^* x_j}{\delta_j} \right) - \frac{1}{2} \sum_{u \in V} \frac{1}{d_u} \sum_{i,j \in E} w_i^{\frac{1}{2}} \left( \vec{B}_{ui}^* \vec{B}_{uj} \frac{x_i^* x_j}{\delta_i^{\frac{1}{2}} \delta_j^{\frac{1}{2}}} + \vec{B}_{uj}^* \vec{B}_{ui} \frac{x_j^* x_i}{\delta_j^{\frac{1}{2}} \delta_i^{\frac{1}{2}}} \right) w_j^{\frac{1}{2}} .$$

After rewriting the second summation in the second term as $\sum_{i,j \in E: u \in i \wedge \in j}$ (this is w.l.o.g. due to the summand being 0 if either $u \notin i$ or $u \notin i$), we compactly rewrite the whole expression as

$$\frac{1}{2} \sum_{u \in V} \frac{1}{d_u} \sum_{i,j \in E: u \in i \wedge u \in j} \left( w_j \frac{x_i^* x_i}{\delta_i} + w_i \frac{x_j^* x_j}{\delta_j} - w_i^{\frac{1}{2}} w_j^{\frac{1}{2}} \vec{B}_{ui}^* \vec{B}_{uj} \frac{x_i^* x_j}{\delta_i^{\frac{1}{2}} \delta_j^{\frac{1}{2}}} - w_i^{\frac{1}{2}} w_j^{\frac{1}{2}} \vec{B}_{uj}^* \vec{B}_{ui} \frac{x_j^* x_i}{\delta_j^{\frac{1}{2}} \delta_i^{\frac{1}{2}}} \right) .$$

Now, we proceed by analyzing the three possible cases for the summand.

Case 1.a: $u \in H(i) \cap H(j) \Leftrightarrow \vec{B}_{ui} = 1, \vec{B}_{uj} = 1$. We have $\vec{B}_{ui}^* \vec{B}_{uj} = \vec{B}_{uj}^* \vec{B}_{ui} = 1$.

Case 1.b: $u \in T(i) \cap T(j) \Leftrightarrow \vec{B}_{ui} = -\mathrm{i}, \vec{B}_{uj} = -\mathrm{i}$. We have $\vec{B}_{ui}^* \vec{B}_{uj} = \vec{B}_{uj}^* \vec{B}_{ui} = (-\mathrm{i})^*(-\mathrm{i}) = (-\mathrm{i})(\mathrm{i}) = 1$.

In both cases, we have:

$$w_j \frac{x_i^* x_i}{\delta_i} + w_i \frac{x_j^* x_j}{\delta_j} - w_i^{\frac{1}{2}} w_j^{\frac{1}{2}} \frac{x_i^* x_j}{\delta_i^{\frac{1}{2}} \delta_j^{\frac{1}{2}}} - w_i^{\frac{1}{2}} w_j^{\frac{1}{2}} \frac{x_j^* x_i}{\delta_j^{\frac{1}{2}} \delta_i^{\frac{1}{2}}} = \left( \frac{w_j^{\frac{1}{2}} x_i}{\delta_i^{\frac{1}{2}}} - \frac{w_i^{\frac{1}{2}} x_j}{\delta_j^{\frac{1}{2}}} \right)^* \left( \frac{w_j^{\frac{1}{2}} x_i}{\delta_i^{\frac{1}{2}}} - \frac{w_i^{\frac{1}{2}} x_j}{\delta_j^{\frac{1}{2}}} \right) .$$

Letting $x_i = a_i + \mathrm{i} b_i$ and $x_j = a_j + \mathrm{i} b_j$, this expression boils down to

$$\left( \frac{w_j^{\frac{1}{2}} a_i}{\delta_i^{\frac{1}{2}}} - \frac{w_i^{\frac{1}{2}} a_j}{\delta_j^{\frac{1}{2}}} \right)^2 + \left( \frac{w_j^{\frac{1}{2}} b_i}{\delta_i^{\frac{1}{2}}} - \frac{w_i^{\frac{1}{2}} b_j}{\delta_j^{\frac{1}{2}}} \right)^2 .$$

Case 2.a: $u \in H(i) \cap T(j) \Leftrightarrow \bar{B}(u,i) = 1, \bar{B}(u,j) = -\mathrm{i}$. We have $\bar{B}(u,i)^* \bar{B}(u,j) = (1)^*(-\mathrm{i}) = -\mathrm{i}$ and $\bar{B}(u,j)^* \bar{B}(u,i) = (-\mathrm{i})^*(1) = \mathrm{i}$. In this case, we have:

$$w_j \frac{x_i^* x_i}{\delta_i} + w_i \frac{x_j^* x_j}{\delta_j} + \mathrm{i} w_i^{\frac{1}{2}} w_j^{\frac{1}{2}} \frac{x_i^* x_j}{\delta_i^{\frac{1}{2}} \delta_j^{\frac{1}{2}}} - \mathrm{i} w_i^{\frac{1}{2}} w_j^{\frac{1}{2}} \frac{x_j^* x_i}{\delta_j^{\frac{1}{2}} \delta_i^{\frac{1}{2}}} .$$

Letting $x_i = a_i + \mathrm{i}b_i$ and $x_j = a_j + \mathrm{i}b_j$, this expression reads

$$\left( \frac{w_j^{\frac{1}{2}} a_i}{\delta_i^{\frac{1}{2}}} - \frac{w_i^{\frac{1}{2}} b_j}{\delta_j^{\frac{1}{2}}} \right)^2 + \left( \frac{w_i^{\frac{1}{2}} a_j}{\delta_j^{\frac{1}{2}}} + \frac{w_j^{\frac{1}{2}} b_i}{\delta_i^{\frac{1}{2}}} \right)^2 .$$

Case 2.b: $u \in T(i) \cap H(j) \Leftrightarrow \bar{B}(u,i) = -\mathrm{i}, \bar{B}(u,j) = 1$. We have $\bar{B}(u,i)^* \bar{B}(u,j) = (-\mathrm{i})^*(1) = \mathrm{i}$ and $\bar{B}(u,j)^* \bar{B}(u,i) = (1)^*(-\mathrm{i}) = -\mathrm{i}$. In this case, we have:

$$w_j \frac{x_i^* x_i}{\delta_i} + w_i \frac{x_j^* x_j}{\delta_j} - \mathrm{i} w_i^{\frac{1}{2}} w_j^{\frac{1}{2}} \frac{x_i^* x_j}{\delta_i^{\frac{1}{2}} \delta_j^{\frac{1}{2}}} + \mathrm{i} w_i^{\frac{1}{2}} w_j^{\frac{1}{2}} \frac{x_j^* x_i}{\delta_j^{\frac{1}{2}} \delta_i^{\frac{1}{2}}} .$$

Letting $x_i = a_i + \mathrm{i}b_i$ and $x_j = a_j + \mathrm{i}b_j$, this latter expression reads

$$\left( \frac{w_j^{\frac{1}{2}} a_i}{\delta_i^{\frac{1}{2}}} + \frac{w_i^{\frac{1}{2}} b_j}{\delta_j^{\frac{1}{2}}} \right)^2 + \left( \frac{w_i^{\frac{1}{2}} a_j}{\delta_j^{\frac{1}{2}}} - \frac{w_j^{\frac{1}{2}} b_i}{\delta_i^{\frac{1}{2}}} \right)^2 .$$

The final equation reported in the statement of the theorem is obtained by combining the four cases we just analyzed. $\qquad\square$

**Corollary 3.** $\vec{\mathbb{L}}_N$ *is positive semidefinite.*

*Proof.* Since $\vec{\mathbb{L}}_N$ is Hermitian, it can be diagonalized as $U\Lambda U^*$ for some $U \in \mathbb{C}^{n \times n}$ and $\Lambda \in \mathbb{R}^{n \times n}$, where $\Lambda$ is diagonal and real. We have $x^* \vec{\mathbb{L}}_N x = x^* U \Lambda U^* x = y^* \Lambda y$ with $y = U^* x$. Since $\Lambda$ is diagonal, we have $y^* \Lambda y = \sum_{u \in V} \lambda_u y_u^2$. Thanks to Theorem 2, the quadratic form $x^* \vec{\mathbb{L}}_N x$ associated with $\vec{\mathbb{L}}_N$ is a sum of squares of real values and, hence, nonnegative. Combined with $x^* \vec{\mathbb{L}}_N x = \sum_{u \in L(V)} \lambda_u y_u^2$, we deduce $\lambda_u \geq 0$ for all $u \in L(V)$, where $L(V)$ is the vertex set of $\mathrm{DLG}(\vec{H})$. $\qquad\square$

**Theorem 4.** $\vec{\mathbb{Q}}_N$ *is positive semidefinite.*

*Proof.* $\vec{\mathbb{Q}}_N$ is positive semidefinite by construction. Indeed, since, by definition, $\vec{\mathbb{Q}}_N = \vec{D}_e^{-\frac{1}{2}} W^{\frac{1}{2}} \vec{B}^* \vec{D}_v^{-1} \vec{B} W^{\frac{1}{2}} \vec{D}_e^{-\frac{1}{2}}$, the quadratic form $x^* \vec{\mathbb{Q}}_N x$ satisfies the relationship $x^* \vec{\mathbb{Q}}_N x = x^* \vec{D}_e^{-\frac{1}{2}} W^{\frac{1}{2}} \vec{B}^* \vec{D}_v^{-1} \vec{B} W^{\frac{1}{2}} \vec{D}_e^{-\frac{1}{2}} x = (\vec{D}_v^{-1} \vec{B} W^{\frac{1}{2}} \vec{D}_e^{-\frac{1}{2}} x)^* (\vec{D}_v^{-1} \vec{B} W^{\frac{1}{2}} \vec{D}_e^{-\frac{1}{2}} x) = ||(\vec{D}_v^{-1} \vec{B} W^{\frac{1}{2}} \vec{D}_e^{-\frac{1}{2}}) x||_2 \geq 0$. $\qquad\square$

**Corollary 5.** $\lambda_{\max}(\vec{\mathbb{L}}_N) \leq 1$ *and* $\lambda_{\max}(\vec{\mathbb{Q}}_N) \leq 1$.

*Proof.* $\lambda_{\max}(\vec{\mathbb{L}}_N) \leq 1$ holds if and only if $\vec{\mathbb{L}}_N - I \preceq 0$. Since $\vec{\mathbb{L}}_N = I - \vec{\mathbb{Q}}_N$ holds by definition, we need to prove $-\vec{\mathbb{Q}}_N \preceq 0$. This is the case due to Theorem 4.

Similarly, $\lambda_{\max}(\vec{\mathbb{Q}}_N) \leq 1$ holds if and only if $\vec{\mathbb{Q}}_N - I \preceq 0$. Since $\vec{\mathbb{Q}}_N = I - \vec{\mathbb{L}}_N$ holds by definition, we need to prove $-\vec{\mathbb{L}}_N \preceq 0$. This is the case due to Theorem 3. $\qquad\square$

**Proposition 6.** *The convolution operator derived from equation 4 by setting $\mathcal{L} = \vec{\mathbb{L}}_N$ with parameters $\theta_0$ and $\theta_1$ is the same as the convolution operator obtained by using $\mathcal{L} = \vec{\mathbb{Q}}_N$ with parameters redefined as $\theta_0' = \theta_0 + \theta_1$ and $\theta_1' = -\theta_1$.*

*Proof.* Consider the two operators $\theta_0 I + \theta_1 \vec{\mathbb{L}}_N$ and $\theta_0' I + \theta_1' \vec{\mathbb{Q}}_N$. Since $\vec{\mathbb{L}}_N = I - \vec{\mathbb{Q}}_N$, the first operator reads: $\theta_0 I + \theta_1 (I - \vec{\mathbb{Q}}_N)$. This is rewritten as $(\theta_0 + \theta_1) I - \theta_1 \vec{\mathbb{Q}}_N$. By operating the choice $\theta_0' = \theta_0 + \theta_1$ and $\theta_1 = -\theta_1'$, the second operator is obtained. $\qquad\square$

## D  COMPLEXITY AND EXPRESSIVENESS OF DLGNET

Let us assume (w.l.o.g.) that each of DLGNet's convolutional layers has $c$ input and output channels, while the last layer has $c$ input and $c'$ output channels ($c'$ is also the number of input channel to the linear layers). Let $d$ be number of output channel of the last linear layer (where $d$ is the number of classes to be predicted). With $\ell$ convolutional layers and $S$ linear layers, DLGNet's complexity is $O(mnc_0) + O(\ell(m^2c + mc^2) + mc + (S-1)(mc'^2) + mc'd + md)$. Assuming $O(c) = O(c') = O(d) = \bar{c}$, we have a complexity of $O(\ell(m^2\bar{c}) + (\ell + S)(m\bar{c}^2))$. This shows that DLGNet has a quadratic complexity w.r.t. the number of hyperpedges $m$ and the asymptotic number of channels $\bar{c}$.

The detailed calculations for the (inference) complexity of DLGNet are as follows.

1. The Directed Line Graph Laplacian $\vec{\mathbb{L}}_N$ is constructed in time $O(m^2n)$, where the factor $n$ is due to the need for computing the product between two columns of $\vec{B}$ (i.e., two rows of $B^*$) to calculate each entry of $\vec{\mathbb{L}}_N$. After $\vec{\mathbb{L}}_{\mathbb{N}}$ has been computed, the convolution matrix $\hat{Y} \in \mathbb{C}^{m \times m}$ is constructed in time $O(m^2)$. Note that such a construction is carried out entirely in pre-processing and is not required at inference time.

2. Constructing the feature matrix $X = \vec{B}^*X'$ requires $O(mnc_0)$ elementary operations.

3. Each of the $\ell$ convolutional layers of DLGNet requires $O(m^2c + mc^2 + mc) = O(m^2c + mc^2)$ elementary operations across 3 steps. Let $X^{l-1}$ be the input matrix to layer $l = 1, \ldots, \ell$. The operations that are carried out are the following ones.

   (a) $\vec{\mathbb{L}}_{\mathbb{N}}$ is multiplied by the hyperedge-feature matrix $X^{l-1} \in \mathbb{C}^{m \times c}$, obtaining $P^{l_1} \in \mathbb{C}^{m \times c}$ in time $O(m^2c)$ (we assume, for simplicity, that matrix multiplications takes cubic time);

   (b) The matrices $P^{l_0} = IX^{l-1} = X^{l-1}$ and $P^{l1}$ are multiplied by the weight matrices $\Theta_0, \Theta_1 \in \mathbb{R}^{c \times c}$ (respectively), obtaining the intermediate matrices $P^{l_{01}}, P^{l_{11}} \in \mathbb{C}^{n \times c}$ in time $O(mc^2)$ .

   (c) The matrices $P^{l_{01}}$ and $P^{l_{11}}$ are additioned in time $O(mc)$ to obtain $P^{l_2}$.

   (d) The activation function $\phi$ is applied component wise to $P^{l_2}$ in time $O(mc)$, resulting in the output matrix $X^l \in \mathbb{C}^{m \times c}$ of the $l$-th convolutional layer.

4. The unwind operator transforms $X^\ell$ (the output of the last convolutional layer $\ell$) into the matrix $U^0 \in \mathbb{R}^{n \times 2c}$ in linear time $O(mc)$.

5. Call $U^{s-1}$ the input matrix to each linear layer of index $s = 1, \ldots, S$. The application of the $s$-th linear layer to $U^{s-1} \in \mathbb{C}^{m \times c'}$ requires multiplying $U^{s-1}$ by a weight matrix $M_s \in \mathbb{C}^{c' \times c'}$ (where $c'$ is the number of channels from which and into which the feature vector of each node is projected). This is done in time $O(mc'^2)$.

6. In the last linear layer of index $S$, the input matrix $U^{S-1} \in \mathbb{R}^{m \times c'}$ is projected into the output matrix $O \in \mathbb{R}^{m \times d}$ in time $O(nc'd)$.

7. The application of the Softmax activation function takes linear time $O(md)$.

We deduce an overall complexity of $O(mnc_0) + O(\ell(m^2c + mc^2) + mc + (S-1)(mc'^2) + mc'd + md)$. Assuming $O(c) = O(c') = O(d) = \bar{c}$, such a complexity coincides with $O(\ell(m^2\bar{c}) + (\ell + S)(m\bar{c}^2))$.

First, notice that our proposed Directed Line Graph coincides with the standard line graph when it is created starting from a hypergraph is 2-uniform and undirected. In such a case, it well-known that, given two unweighted graphs $G_1, G_2$ and their line graphs $L(G_1), L(G_2)$, it can either be that $G_1 \equiv_{WL} G_2$ while $L(G_1) \not\equiv_{WL} L(G_2)$ or viceversa, where $\equiv_{WL}$ means "equivalent under the Weisfeiler-Leman test of the first order". Therefore, even in the simplest case where the hypergraph is unweighted and 2-uniform, there are graphs where the expressive power of DLGNet is higher than that of a standard GCN as well as cases where its power is lower. Clearly, if we assume that the line graph (and not the original graph) is the input to DLGNet, then DLGNet is at least as powerful as the classical GCN due to the fact that their Laplacian operators, and thus their convolutional layers, coincide.

## E  FURTHER DETAILS ON THE DATASETS

Details on the datasets composition are reported in Tables 3, 4, 5. Most of the elements of `Dataset-1` belong to the first two classes, which concern the addition of functional groups to a chemical compound: alkyl and aryl groups for *Class 1* and acyl groups for *Class 2*, comprising more than 17K species. Less populated classes involve specific chemical transformations, such as *Class 3* (C–C bond formation) which contains less than 1000 elements.

`Dataset-2` presents solely three classes. The elements of the first class (C–C bond formation) are extracted from two separate collections present in the Open Reaction Database (ORD) Project (Kearnes et al., 2021). Those are the Reizman et al. (2016) data for the Pd-catalyzed Su-zuki–Miyaura cross-coupling reactions and a vast collection of Pd-catalyzed imidazole-aryl coup-ling reactions, via C-H arylation. The elements of *Class 2* (N-arylation) includes data of Pd-catalyzed N-arylation (Buchwald-Hartwig) reactions from the AstraZeneca ELN dataset, also gen-erated from the ORD website. This class has been further divided in 3 sub-classes according to the nature of the aryl halide used for the coupling. Finally, the third class contains an ORD collection of data for amide bond formation processes. We have been able to extract sub-categories from two of them. Those are *Class 1* (C–C bond formation) and *Class 2* (N-arylation processes) and contain two and three sub-classes, respectively. The most populated class is Imidazole-aryl coupling, comprising around 1500 elements belonging to the class of palladium-catalyzed imidazole C-H arylation. The chemical diversity in this class is ensured by the use of 8 aryl bromides and 8 imidazole compounds. Furthermore, in terms of reaction conditions, the collection presents 24 different monophosphine ligands.

Unlike the previous ones, `Dataset-3` has been assembled starting from competitive processes; therefore it contains almost the same amount of elements ($\sim 300$) for the two classes: Bimolecular nucleophilic substitution ($S_N2$) and eliminations (E2). The reactants–which are in common between $S_N2$ and E2—are substituted alkane compounds and nucleophile agents. The substituents span a range of electron donating and electron withdrawing effect strengths, including methyl, cyano, amine, and nitro functional groups. The nucleophiles have been chosen either between halide or hydrogen anions, while the molecular skeleton is ethane.

In Table 6, we present statistics on the Hypergraphs and Directed Line Graphs derived from the three datasets. As shown in the table, it is highly unlikely that the graph convolutional layer reduces to a linear layer due to the lack of neighboring nodes. On the contrary, each node is likely to have neighboring connections.

Table 3: Distribution of the reactions in the `Dataset-1`.

| Reaction class | Reaction name | Num Reactions |
|---|---|---|
| 1 | Heteroatom alkylation and arylation | 15151 |
| 2 | Acylation and related process | 11896 |
| 3 | C-C bond formation | 909 |
| 4 | Heterocycle formation | 4614 |
| 5 | Protections | 1834 |
| 6 | Deprotections | 5662 |
| 7 | Reductions | 672 |
| 8 | Oxidations | 811 |
| 9 | Functional group interconversion | 8237 |
| 10 | Functional group addition | 230 |

## F  FURTHER DETAILS ON THE EXPERIMENTS

**Hardware.**  The experiments were conducted on 2 different machines:

1. An Intel(R) Xeon(R) Gold 6326 CPU @ 2.90GHz with 380 GB RAM, equipped with an NVIDIA Ampere A100 40GB.

Table 4: Distribution of the reactions in the `Dataset-2`.

| Reaction class | Reaction name | Num Reactions |
|---|---|---|
| 1 | C-C bond formation | 1921 |
| | - Reizman Suzuki Cross-Coupling | 385 |
| | - Imidazole-aryl coupling | 1536 |
| 2 | Heteroatom (N) arylation: | 657 |
| | - Amine + Aryl bromide | 278 |
| | - Amine + Aryl chloride | 299 |
| | - Amine + Aryl iodide | 80 |
| 3 | Amide bond formation | 960 |

Table 5: Distribution of reactions in the `Dataset-3`.

| Reaction class | Reaction name | Num Reactions |
|---|---|---|
| 1 | Bimolecular nucleophilic substitution ($S_N2$) | 301 |
| 2 | Bimolecular elimination (E2) | 348 |

Table 6: Statistics on Hypergraphs and Directed Line Graphs

| Dataset | $n$ | $|E|$ | Density | Vertex Degree |
|---|---|---|---|---|
| `Dataset-1` | 100523 | 50016 | 0.10% | 48.05 |
| `Dataset-2` | 956 | 3021 | 43.29% | 1036.32 |
| `Dataset-3` | 670 | 649 | 73.03% | 511.20 |

2. A 12th Gen Intel(R) Core(TM) i9-12900KF CPU @ 3.20GHz CPU with 64 GB RAM, equipped with an NVIDIA RTX 4090 GPU.

**Model Settings.**    We trained every learning model considered in this paper for up to 1000 epochs. We adopted a learning rate of $5 \cdot 10^{-3}$ and employed the optimization algorithm Adam with weight decays equal to $5 \cdot 10^{-4}$ (in order to avoid overfitting). We set the number of linear layers to 2, i.e. $\ell = 2$, for all the models.

We adopted a hyperparameter optimization procedure to identify the best set of parameters for each model. In particular, the hyperparameter values are:

- For AllDeepSets and ED-HNN, the number of basic block is chosen in $\{1, 2, 4, 8\}$, the number of MLPs per block in $\{1, 2\}$, the dimension of the hidden MLP (i.e., the number of filters) in $\{64, 128, 256, 512\}$, and the classifier hidden dimension in $\{64, 128, 256\}$.

- For AllSetTransformer the number of basic block is chosen in $\{2, 4, 8\}$, the number of MLPs per block in $\{1, 2\}$, the dimension of the hidden MLP in $\{64, 128, 256, 512\}$, the classifier hidden dimension in $\{64, 128, 256\}$, and the number of heads in $\{1, 4, 8\}$.

- For UniGCNII, HGNN, HNHN, HCHA/HGNN$^+$, LEGCN, and HCHA with the attention mechanism, the number of basic blocks is chosen in $\{2, 4, 8\}$ and the hidden dimension of the MLP layer in $\{64, 128, 256, 512\}$.

- For HyperGCN, the number of basic blocks is chosen in $\{2, 4, 8\}$.

- For HyperND, the classifier hidden dimension is chosen in $\{64, 128, 256\}$.

- For PhenomNN, the number of basic blocks is chosen in $\{2, 4, 8\}$. We select four different settings:

    1. $\lambda_0 = 0.1$, $\lambda_1 = 0.1$ and prop step$= 8$,
    2. $\lambda_0 = 0$, $\lambda_1 = 50$ and prop step$= 16$,
    3. $\lambda_0 = 1$, $\lambda_1 = 1$ and prop step$= 16$,
    4. $\lambda_0 = 0$, $\lambda_1 = 20$ and prop step$= 16$.

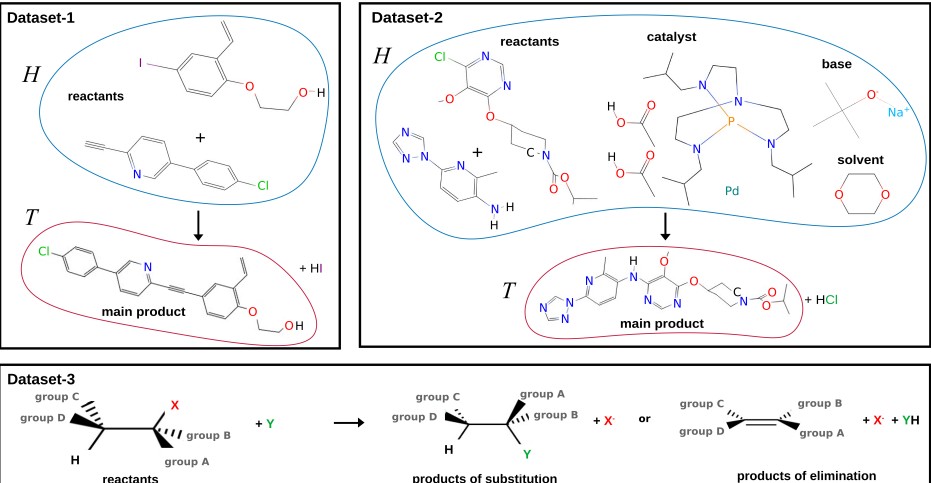

Figure 3: **(Upper panel, left)**: example from `Dataset-1`. C–C bond formation via reaction of alkyne with alkyl halide; only bi-molecular reactant and main product are taken into account (any byproduct is omitted). **(Upper panel, right)**: example from `Dataset-2`. C–N bond formation via Buchwald-Hartwig amination; apart from bi-molecular reactant (amine and aryl halide) and main product, catalyst (palladium compound), solvent (dioxane) and base (sodium tert-butoxide) structures are also present. Chemical elements: carbon (C), nitrogen (N), oxygen (O), hydrogen (H), chlorine (Cl), iodine (I), sodium (Na), phosphorus (P) and palladium (Pd). Single, double and triple black lines: bonds between C atoms. $H$, $T$: Head and Tail of the directed hypergraph. **(Lower panel)**: schematic representation of `Dataset-3` elements. Left side: reactants; right side: competitive outcomes between bimolecular nucleophilic substitution ($S_N2$) or bimolecular elimination (E2). Thus, each element is composed either of a bi-molecular reactant and a bi-molecular product ($S_N2$ class), or a bi-molecular reactant and a tri-molecular product (E2 class). X and Y: leaving group and nucleophile agent. Groups A-D: different substituents attached to the alkane carbon backbone (black).

- For DHM and DHRL, the classifier hidden dimension is chosen in $\{64, 128, 256\}$.
- For DLGNet, the number of convolutional layers is chosen in $\{1, 2, 3\}$, the number of filters in $\{64, 128, 256, 512\}$, and the classifier hidden dimension in $\{64, 128, 256\}$. We tested DLGNet both with the input feature matrix $X \in \mathbb{C}^{n \times c}$ where $\Re(X) = \Im(X) \neq 0$ and with $\Im(X) = 0$.

**How to Transfer The Features.** As mention in Section 4, a key aspect of our approach involves transferring features from the nodes of the hypergraph to their corresponding hyperedges, i.e., the nodes of the directed line graph. To clarify this mechanism, we provide a simple example. Consider a directed hypergraph $\vec{H} = (V, E)$, where the vertex set is $V = \{u, v, c\}$ and the hyperedge set consists of $E = \{e_1\}$. In $\vec{H}$, we have $H(e_1) = \{u, v\}$ and $T(e_1) = \{c\}$. Each vertex is assigned a feature vector $x'_u, x'_v, x'_c = 1$ and the hyperedge has a unit weight, i.e. $w_{e_1} = 1$. Recalling that $X = B^* X'$, the feature vector $x_1$ of the hyperedge $e_1$ is then calculated as:

$$x_1 = \vec{B}^*_{1u} \cdot x_u + \vec{B}^*_{1v} \cdot x_v + \vec{B}^*_{1c} \cdot x_c = 2 + i.$$

In the case where $\vec{H} = H$, i.e., when the hypergraph is undirected, we have $\vec{B}^* = B^\top$. The feature vector $x_1$ of the hyperedge $e_1$ is then calculated as:

$$x_1 = B_{1u} \cdot x_u + B_{1v} \cdot x_v + B_{1c} \cdot x_c = 3.$$

As illustrated by this example, in the specific case of a directed line graph, the feature vector can feature both real and imaginary components, depending on the topology of the hypergraph encoded by $\vec{B}$.

**Training times** An overview of the training times for the various architectures considered in this paper is reported in Table 7.

Table 7: Comparison of the training times for various graph-based learning methods, also including their number of parameters.

| Method | # Params | Avg Time (s) | Std Time (s) | Avg Time (min:s) |
|---|---|---|---|---|
| HGNN | 22,346 | 72.66 | 1.32 | 1min 12.66s |
| HCHA/HGNN$^+$ | 44,682 | 77.64 | 1.21 | 1min 17.64s |
| HCHA w/ Attention | 179,742 | 141.50 | 0.54 | 2min 21.50s |
| HNHN | 4,118,538 | 502.68 | 0.53 | 8min 22.68s |
| UniGCNII | 175,754 | 134.49 | 0.50 | 2min 14.49s |
| AllDeepSets | 4,200,750 | 436.68 | 0.50 | 7min 16.68s |
| AllSetTransformer | 2,196,106 | 344.55 | 0.45 | 5min 44.55s |
| ED-HNN | 2,052,426 | 263.91 | 0.46 | 4min 23.91s |
| Phenom | 121,098 | 335.79 | 0.33 | 5min 35.79s |
| DHM | 591,114 | 112.52 | 0.67 | 1min 52.52s |
| DHRL | 262,218 | 104.99 | 0.64 | 1min 44.99s |
| DGLNet | 3,264,330 | 383.54 | 0.50 | 6min 23.54s |

## G DIRECTED LINE GRAPH LAPLACIAN AND THE OTHER LAPLACIANS.

The Laplacian proposed in this work differs in several key aspects from existing Laplacians designed to handle both directed and undirected edges in graphs, such as the *Magnetic Laplacian* (Lieb &Loss, 1993) and the *Sign-Magnetic Laplacian* (Fiorini et al., 2023). Specifically, our approach constructs the Laplacian from a complex matrix $A(DLG(\vec{H}))$, as defined in Equation equation 8. The difference also extends to behavior: the Directed Line Graph Laplacian exhibits a unique characteristic, as both its real and imaginary components can be nonzero simultaneously, as can be seen through Equation equation 10. This is different from the case of the *Sign-Magnetic Laplacian*, which can only have one of the two components different from zero at any given time, and also from the case of the *Magnetic Laplacian*, which coincides with the *Sign Magnetic Laplacian* when $q = \frac{1}{4}$ and the graph has binary weights. Let us note that the *Magnetic Laplacian* can also have both components different from zero, but such a behavior is influenced by both the edge weight and the value of $q$, and may lead to the sign-pattern inconsistency described in Fiorini et al. (2023), which our proposed Directed Line Graph Laplacian does not suffer from.

## H CONFUSION MATRIX

We report the confusion matrices of `Dataset-1` in Figure 4 and `Dataset-2` in Figure 5. We can extract some insights from these two matrices, in particular:

- `Dataset-1`. DLGNet achieves a maximum performance of 88% in classifying the *Class 5* (Protections). However, its performance drops for *Class 4* and *Class 9* (Heterocycle formations and Functional group interconversions), where it correctly predicts only 44% and 41%, respectively.

- `Dataset-2`. DLGNet accurately classifies the sub-classes relative to the C–C bond formations (Reizman Suzuki Cross-Coupling and Imidazole-aryl coupling), as well as the Amide bond formations. On the other hand, the remaining three N-arylation sub-classes are poorly discriminated. This behavior can likely be attributed to the fact that the former are derived from different collections of Pd-catalyzed cross-coupling reactions, each exhibiting distinct features in terms of participant molecules (e.g. imidazole compounds). In contrast, all of the elements in the N-arylation classes share the same reaction mechanism (Buchwald-Hartwig amination); this poses a greater challenge, which results in decreased accuracy when predicting the correct class.

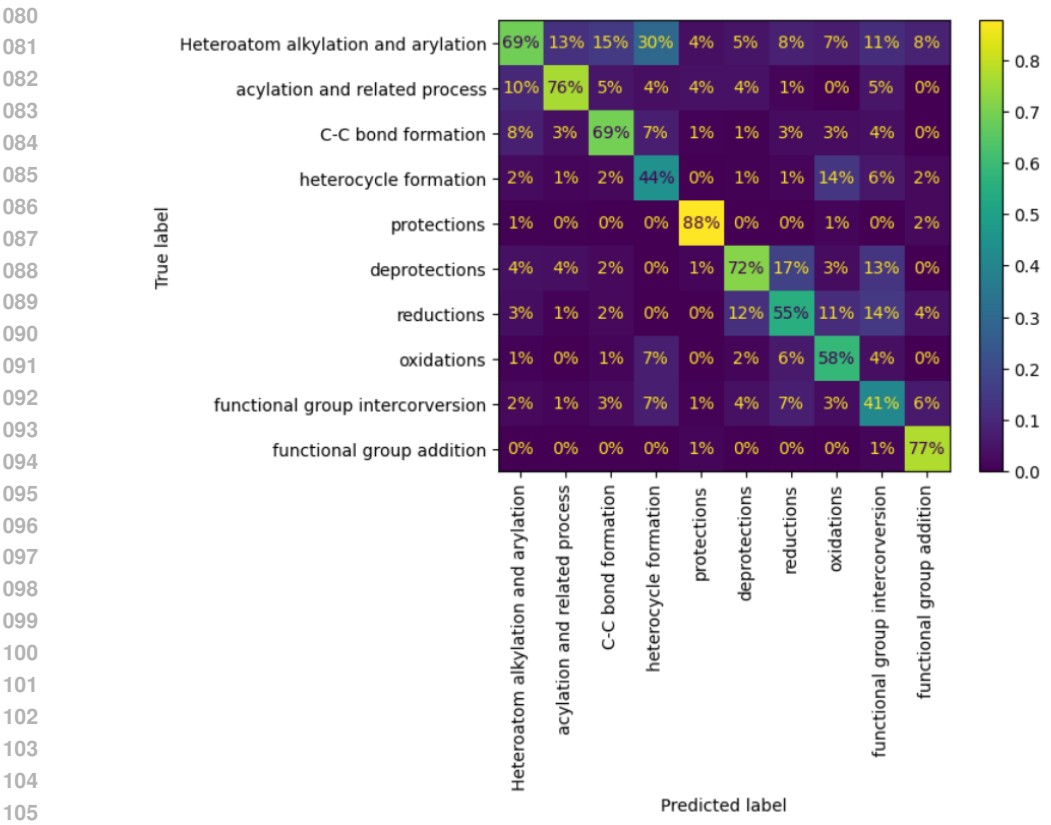

Figure 4: `Dataset-1` confusion matrix.

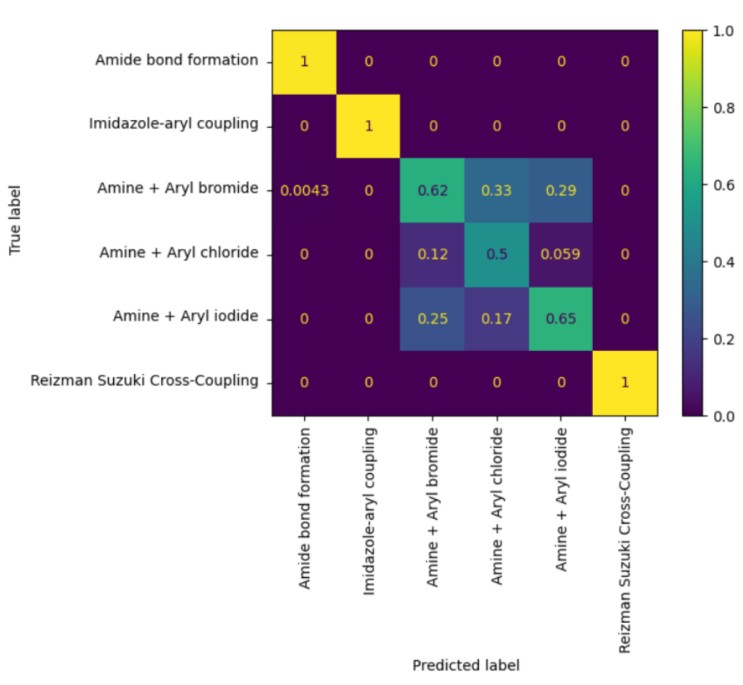

Figure 5: `Dataset-2` confusion matrix.

# I FROM A DIRECTED HYPERGRAPH TO THE DIRECTED LINE GRAPH LAPLACIAN

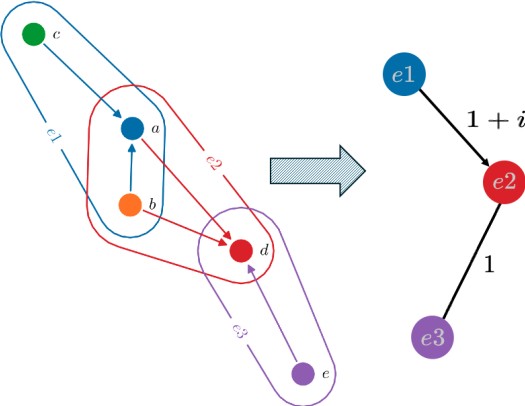

Figure 6: An example illustrating the transformation of a hypergraph (left) into its corresponding directed line graph (right).

To illustrate the construction of the directed line graph and the associated Directed Line Graph Laplacian, consider a directed hypergraph $\vec{H} = (V, E)$ where the vertex set is $V = \{a, b, c, d, e\}$ and the hyperedge set is $E = \{e_1, e_2, e_3\}$. The incidence relationships are defined as follows:

- $H(e_1) = \{b, c\}, T(e_1) = \{a\}$,
- $H(e_2) = \{a, b\}, T(e_2) = \{d\}$,
- $H(e_3) = \{e\}, T(e_3) = \{d\}$.

Each hyperedge is assigned a unit weight (i.e., $W = I$). The cardinalities (densities) of the hyperedges are $\delta_{e_1} = 3$, $\delta_{e_2} = 2$, and $\delta_{e_3} = 2$.

We construct $DLG(\vec{H})$ using the following matrices: the incidence matrix $\vec{B}$, its conjugate transpose $\vec{B}^*$, the vertex degree matrix $D_v$, and the hyperedge degree matrix $D_e$. The incidence matrix $\vec{B}$ and its conjugate transpose are:

$$\vec{B} = \begin{bmatrix} -i & 1 & 0 \\ 1 & 1 & 0 \\ 1 & 0 & 0 \\ 0 & -i & -i \\ 0 & 0 & 1 \end{bmatrix} \quad \vec{B}^* = \begin{bmatrix} i & 1 & 1 & 0 & 0 \\ 1 & 1 & 0 & i & 0 \\ 0 & 0 & 0 & i & 1 \end{bmatrix}.$$

The vertex degree matrix $D_v$ and the hyperedge degree matrix $D_e$ are given by:

$$D_v = \begin{bmatrix} 2 & 0 & 0 & 0 & 0 \\ 0 & 2 & 0 & 0 & 0 \\ 0 & 0 & 1 & 0 & 0 \\ 0 & 0 & 0 & 2 & 0 \\ 0 & 0 & 0 & 0 & 1 \end{bmatrix} \quad D_e = \begin{bmatrix} 3 & 0 & 0 \\ 0 & 3 & 0 \\ 0 & 0 & 2 \end{bmatrix}.$$

Using these matrices, the adjacency matrix $A$ of the directed line graph $DLG(\vec{H})$ is:

$$A = \vec{B}^*\vec{B} - D_e = \begin{bmatrix} 0 & 1+i & 0 \\ 1-i & 0 & 1 \\ 0 & 1 & 0 \end{bmatrix}. \tag{14}$$

By Definition 1, the directed line graph $DLG(\vec{H})$ has three vertices, corresponding to the hyperedges $e_1$, $e_2$, and $e_3$ of the original hypergraph $\vec{H}$. An edge exists between two vertices in $DLG(\vec{H})$ if and only if their corresponding hyperedges in $\vec{H}$ are incident. In the specific example (illustrated in Figure 6), DLG($\vec{H}$) contains two edges, whose direction and weight are determined by the adjacency matrix $A$ (in equation 14). Without loss of generality, we consider the upper triangular part of $A$ to assign weights to the edges and define the directions: In the example considered, one edge will be directed and have a weight equal to $1+i$ (i.e., $e_1 \stackrel{1+i}{\rightarrow} e_2$), while the other edge will be undirected and have a weight equal to $1$ ($e_2 \stackrel{1}{\rightharpoonup} e_3$).

Using the equation 9. We can calculate the proposed Directed Line Graph Laplacian $\vec{\mathbb{L}}_N$ as follows:

$$\vec{\mathbb{L}}_N = I - \vec{\mathbb{Q}}_N := \vec{D}_e^{-\frac{1}{2}} \vec{B}^* \vec{D}_v^{-1} \vec{B} \vec{D}_e^{-\frac{1}{2}} = \begin{bmatrix} 0.333 & -0.167 - 167i & 0 \\ -0.167 + 0.167i & 0.5 & -0.204 \\ 0 & -0.204 & 0.25 \end{bmatrix}.$$

By inspecting $\vec{\mathbb{L}}_{\mathbb{N}}$, one can observe that it encodes the elements of the hypergraph $\vec{H}$ in the following way:

1. The real components of off-diagonal entries in $\vec{\mathbb{L}}_{\mathbb{N}}$ encode the fact that, in the underlying hypergraph $\vec{H}$, the vertex belongs to the head set or tail set simultaneously in two different hyperedges. For example, $\vec{\mathbb{L}}_{\mathbb{N}}(2,3) = -0.204$ indicates that $H(e_2) \cap H(e_3) \neq \emptyset$ or $T(e_2) \cap T(e_3) \neq \emptyset$. In this specific case, $T(e_2) \cap T(e_3) = \{d\}$. Similarly, $\Re(\vec{\mathbb{L}}_{\mathbb{N}}(1,2)) = -0.167$ arises from the fact that $e_1$ and $e_2$ share the vertex $b$ in their head sets.

2. The imaginary component captures the hyperedge directionality based on the underlying hypergraph $\vec{H}$, where a node belongs to the head set of one hyperedge and the tail set of another. For example, $\Im(\vec{\mathbb{L}}_N(1,2)) = -\Im(\vec{\mathbb{L}}_N(2,1)) = -0.167$, indicating that $a \in T(e_1) \cap H(e_2)$.

3. The absence of any relationships between hyperedges $e1$ and $e3$ is encoded by $0$ in $DGL(\vec{H})$. Specifically, $\vec{\mathbb{L}}_N(1,3) = \vec{\mathbb{L}}_N(3,1) = 0$.

4. The *self-loop information* (a measure of how strongly the feature of a vertex depends on its current value within the convolution operator) is encoded by the diagonal of $\vec{\mathbb{L}}_N$.

## J  OUR INCIDENT MATRIX $\vec{B}$

Utilizing our incidence matrix $\vec{B}$, where $\vec{B}_{ve} = -i$ if $v \in T(e)$, with complex numbers allows us to encode directionality and construct a Laplacian, the Directed Line Graph Laplacian, that is Hermitian and meets the necessary properties for applying a spectral-based approach.

If, instead, we had chosen to use $B_{ve} = -1$ if $v \in T(e)$, we would have lost the directionality of the hypergraph. To illustrate, consider (for simplicity—this can be observed also for more hypergraphs) a graph with nodes $1, 2$ and edges $e_1 = (2, 1)$ and $e_2 = (3, 2)$.

We have $\vec{B} = \begin{pmatrix} -i & 0 \\ 1 & -i \\ 0 & 1 \end{pmatrix}$ and $B = \begin{pmatrix} -1 & 0 \\ 1 & -1 \\ 0 & 1 \end{pmatrix}$.

The Laplacian matrix we use in DLGNet is $\vec{L} = \begin{pmatrix} 2 & -i \\ i & 2 \end{pmatrix}$. The Laplacian matrix using the $B$ the reviewer suggests reads $L = \begin{pmatrix} 2 & -1 \\ 1 & 2 \end{pmatrix}$.

Let us recall that the nodes of both matrices correspond to the edges of the graph. Therefore, $\vec{L}_{12}$ indicates the presence of the directed line graph edge $(e_1, e_2)$, which captures the topology of the original graph where edge $e_2$ is seen before edge $e_1$ in a path from node 3 to node 1.

Differently, since $L_{12} = L_{21}$, in $L$ such a directional information is completely lost.

Regarding the solution proposed in (Ma et al., 2024), this approach defines two separate incidence matrices: one for tail elements $B_T = \begin{pmatrix} 1 & 0 \\ 0 & 1 \\ 0 & 0 \end{pmatrix}$ and one for head elements and $B_H = \begin{pmatrix} 0 & 0 \\ 1 & 0 \\ 0 & 1 \end{pmatrix}$.

The Laplacian matrix using $B_T$ and $B_H$ reads $L = \begin{pmatrix} 0 & 0 \\ 1 & 0 \end{pmatrix}$.

This matrix $L$ is not symmetric, which does not fit the framework in which we operate in our paper, in which we are designing a spectral-based convolutional operator. This type of GNN does not permit the use of a non-symmetric matrix, as it requires an eigenvalue decomposition of the Laplacian matrix with real eigenvalues. Thanks to the adoption of a complex-valued $\vec{B}$, our proposed Laplacian matrix is Hermitian and, therefore, admits the required eigenvalue decomposition.

## LLM USAGE STATEMENT

We did *not* use large language models (LLMs) for deriving, checking, or producing any proofs or theoretical results in this paper. All theorems and proofs were conceived, implemented, and validated by the authors.

LLMs were used only as general-purpose assistants for: (i) light prose edits (clarity/grammar) and (ii) minor LaTeX refactoring (e.g., formatting environments). All such edits were manually reviewed. No human-subject data, personally identifiable information, or proprietary datasets were provided to any LLM, and all experimental code runs independently of LLM services.

## REPRODUCIBILITY STATEMENT

We took several steps to support reproducibility. All model components, including the Directed Line Graph Laplacian, training objectives, and update rules, are fully specified in the main text, with additional implementation details in Appendix F. Dataset sources and preprocessing are documented in Appendix E. We report splits, evaluation protocols, and hyperparameter search spaces in Section 5 and Appendix F, and we include hardware information and training schedules.

An anonymized repository with code and scripts to reproduce all tables (including random seeds and configuration files) accompanies this submission (Appendix A). After publication, we will release the non-anonymized repository under the same license. Note that exact bitwise determinism can depend on backend/library settings (e.g., CUDA), but we fix seeds and document any sources of nondeterminism.

## ETHICS STATEMENT

We adhere to the ICLR Code of Ethics. This work studies the Directed Line Graph Laplacian and does not involve human subjects, personally identifiable information, or sensitive attributes. We release an anonymized code repository under a permissive license to facilitate verification and reuse.

As with any graph-learning technique, downstream applications to human-centered data could raise concerns around privacy, fairness, or surveillance. Our contribution is methodological and evaluated on public; nevertheless, we encourage practitioners to assess domain-specific risks, follow applicable regulations, and adopt appropriate safeguards (e.g., data minimization, bias checks) when deploying such models.

