# OpenReview forum: "DLGNet: Hyperedge Classification via a Directed Line Graph for Chemical Reactions"
_ICLR.cc/2026/Conference — Submitted to ICLR 2026_

### Official Review · Reviewer_ZxkP · 2025-10-21

**Soundness:** 3
**Presentation:** 3
**Contribution:** 2
**Rating:** 4
**Confidence:** 4

**Summary:**

This paper presents a graph neural network (GNN) for classifying chemical reactions. The method models reactions as directed hypergraphs and then transforms them into a directed line graph where each vertex represents an entire reaction. A complex-valued Laplacian matrix captures the reaction's directionality, enabling a convolution-like operator for the resulting GNN. Experiments demonstrate superior performance and confirm that encoding this directionality is key to the model's success.

**Strengths:**

1. The paper provides mathematical proofs for the key properties of the proposed Laplacian, i.e., that it is Hermitian and positive semidefinite. This rigour ensures that the method is not an ad-hoc heuristic but a well-defined operator.
2. By isolating the effect of directionality ("DLGNet w/o dir" in Table 2), the paper provides conclusive evidence for its central claim. The blation study confirms that encoding direction is not just helpful but critical for success on this task.

**Weaknesses:**

1. The experiments do not provide evidence that a (hyper)graph-based representation is inherently superior to a simpler, non-structural approach for this specific task. Modern architectures like transformers have shown great success on set-based data. The paper can be strengthened by including a comparison against a strong baseline that does not rely on an explicit graph structure. For example, a Deep Sets model or a Transformer encoder that treats the reactants and products as two distinct sets of molecular fingerprints. Note that the AllDeepSets and AllSetTransformer models use message-passing-style aggregators typical of graph neural networks [1].
2. All tasks are reaction-type classification. There’s no evidence the operator helps on other directed-hypergraph problems on non-chemical domains, e.g., citation co-author network with authors as nodes and citation links between author collaborations (papers) as directed hyperedges.

[1] You are AllSet: A Multiset Function Framework for Hypergraph Neural Networks, ICLR 2022.

**Questions:**

1. Would an expanded ablation on architectural hyperparameters (number of convolutional layers, channel widths, classifier depth) clarify sensitivity and stability?
2. For the merged Dataset-2, what exact deduplication and leakage checks were performed across train/validation/test?
3. In addition to the standardised feature-transfer operator, can baselines be reported with their strongest native hyperedge readouts (e.g., attention-based or learned set pooling) to ensure fair comparison?
4. Beyond reaction-type classification, can the operator be demonstrated on an additional directed-hypergraph task (e.g., hyperedge link prediction) and/or a non-chemical dataset to establish broader applicability?
5. The aggregation of molecular features into reaction features is a critical step. The current method uses a weighted summation. Could the authors elaborate on the rationale for choosing summation over other aggregation functions like mean or max pooling? A brief justification or an ablation study on this design choice would be insightful.
6. Appendix G valuably distinguishes the proposed Laplacian from the Magnetic Laplacian, noting that the DLG Laplacian avoids 'sign-pattern inconsistency.' Could the authors expand on how this theoretical advantage translates into practical benefits, such as improved model stability, expressivity, or performance, specifically for the task of chemical reaction classification?

---

### Official Review · Reviewer_xuyY · 2025-10-30

**Soundness:** 3
**Presentation:** 3
**Contribution:** 3
**Rating:** 8
**Confidence:** 2

**Summary:**

This paper focuses on predicting chemical reactions and introduces a novel method that encodes reaction graphs into a Directed Line Graph (DLG). In this data structure, the triplet set in the raw graph is transformed into a new vertex set. The authors provide theorems for the Signless Directed Line-Graph and the Directed Line Graph Laplacian of the DLG. Additionally, authors building a spectral-based Graph Neural Network to capture features. Experimental results on three real-world chemical reaction datasets demonstrate that DLGNet consistently outperforms all baseline competitors.

**Strengths:**

1. The idea of transformation from standard graph to a hypergraph is interesting and rational for reaction prediction.
2. The  theorems for the Signless Directed Line-Graph and the Directed Line Graph Laplacian of the DLG appears to be comprehensive.

**Weaknesses:**

1. Please remind that the anonymous repository has expired.
2. In the experiments, can the classes that DLGNet easily confuses reveal potential reaction patterns, such as similar regions representing the dominant structures in reactions, reflecting the main areas where bonds are formed and broken?
3. I think an important contribution of this paper is using the Laplacian matrix as prior knowledge for the adjacency matrix in GCN. One thing I'm curious about is whether the Laplacian matrix, after being extracted, can be transformed into the adjacency matrix of the original reaction graph, or integrated as a signal into the original adjacency matrix. For example, $ B * L * B^{-1}$. This is because I feel that DLG compresses too much information, which may affect the upper limit of the neural network's performance.

**Questions:**

See weaknesses.

---

### Official Review · Reviewer_yNwD · 2025-10-31

**Soundness:** 3
**Presentation:** 3
**Contribution:** 1
**Rating:** 2
**Confidence:** 5

**Summary:**

This paper proposes DLGNet as a novel approach to solving the reaction classification task. The key idea is to represent a set of chemical reactions as a hypergraph H, then convert it into a directed line graph (DLG), where each vertex corresponds to a hyperedge in H, and edges are drawn between vertices if their corresponding hyperedges share at least one node in H. DLGNet processes this directed line graph using a spectral GNN. Experimental results show that DLGNet outperforms various baseline methods designed for hypergraph processing across three reaction classification datasets.

**Strengths:**

- The paper clearly presents the often-confusing concepts of hypergraphs and their transformation into line graphs.
- It proposes an extension of the hypergraph incidence matrix from binary {1, 0} values to ternary {1, –i, 0}, and introduces a corresponding Laplacian for the line graph, providing a theoretical foundation for the relationship between the hypergraph and its line graph.
- Using the Laplacian defined above, the authors design a spectral GNN (i.e. DLGNet) and demonstrate its empirical superiority over other hypergraph-based methods on the reaction classification task.

**Weaknesses:**

- If the goal is reaction classification, the selection of baseline methods seems unfair. The paper only compares against hypergraph-based approaches, many of which show very low F1 scores, making the proposed method appear more effective than it might actually be. Reaction classification can also be addressed using traditional chemoinformatics methods and various GNN- or Transformer-based approaches. At the very least, the method should be compared against typical baselines such as a classic ReactionFP [1] and something like [2]-[4]. Without such comparisons, it's hard to assess the real value of the proposed approach.
- There are multiple ways to convert a hypergraph into a graph, not just the line graph approach, but also methods like clique expansion. As shown in cited Zhou et al., NIPS 2006 “Learning with Hypergraphs,” it's also possible to define an adjacency matrix directly from the hypergraph's incidence matrix. This would also be consistent with hypergraph Laplacian theory, and it’s not entirely clear how or why the proposed method is better compared to any potential alternatives.
- I think it would be helpful to include a more concrete explanation of how the hypergraph-based methods being compared differ from the proposed approach. Rather than reducing a hypergraph to a regular graph, it's possible to define message passing directly on the hypergraph and perform prediction by applying global pooling over each hyperedge. In fact, this may allow for more flexible network designs (for example, by incorporating graph transformer layers). However, the paper lacks a clear discussion of how the proposed method compares to or improves upon such approaches, and what specific advantages it offers.

[1] ReactionFP https://doi.org/10.1021/ci5006614
[2] Mapping the space of chemical reactions using attention-based neural networks. (Nat Mach Intell, 2021)  https://doi.org/10.1038/s42256-020-00284-w
[3] Chemical-Reaction-Aware Molecule Representation Learning (ICLR 2022) https://openreview.net/forum?id=6sh3pIzKS-
[4] Rxn Hypergraph: a Hypergraph Attention Model for Chemical Reaction Representation https://arxiv.org/abs/2201.01196

**Questions:**

- It’s unclear whether the primary goal of the paper is to advance ML methods for hypergraphs, to improve reaction classification performance, or both. However, the manuscript frames the work around the reaction classification task and evaluates it solely in that context. If that’s the case, why were the comparisons limited to hypergraph-based methods? Why not include standard approaches commonly used for reaction classification?
- Assuming the transformation into a line graph is a key component, the ternary {1, –i, 0} incidence matrix introduced in Eq. (7) is a very interesting idea. Is his incidence matrix itself the novel contribution of the paper? Or is this idea already exists, and the novelty lies instead in how it’s used to analyze the relationship between hypergraphs and directed line graphs, or in the design of the corresponding Laplacian and spectral GNN? Clarifying this would help position the paper’s contribution more clearly.
- In the end, it's not entirely clear why reducing a hypergraph to a line graph and applying specialized spectral message passing would offer practical advantages over directly designing message passing on the hypergraph itself. Could you provide more clarification on the pros and cons of this line graph reduction approach, especially in comparison to more typical hypergraph-based methods used in your comparisons?

---

### Meta-Review · Area_Chair_GeFC · 2026-01-06

**Summary:**

DLGNet proposes a directed line-graph (DLG) transformation for directed hypergraphs derived from chemical reactions, then performs spectral message passing on the DLG using a novel complex-valued Hermitian Directed Line Graph Laplacian. Across reviewers, there is agreement that the paper is mathematically careful and shows strong empirical performance against the provided hypergraph baselines on three reaction classification datasets. However, the discussions surfaces a high-stakes experimental gap that directly undermines the central empirical claim: the evaluation is not convincingly benchmarked against standard, strong reaction-classification baselines.

As framed, the paper is an application-first submission, but it primarily compares to hypergraph-processing baselines that may be weak for this task. Both Reviewer yNwD (rating 2, confidence 5) and Reviewer ZxkP (rating 4, confidence 4) explicitly question whether the graph/hypergraph formulation is necessary or better than simpler set/transformer approaches, and call for stronger non-graph baselines and fairness checks. With no author response or discussion in the forum to address these points, I cannot reliably assess the true practical advantage of DLGNet for reaction classification, and the current evidence is insufficient for acceptance at ICLR.

**Reviewer Concerns:**

## Addressed (or partially mitigated) within the current submission

- Mathematical soundness of the operator. ZxkP notes the Laplacian is proven Hermitian and PSD, which makes the convolution operator well-defined rather than heuristic. This supports soundness.
- Directionality matters empirically. ZxkP highlights that the ablation “DLGNet w/o dir” offers evidence that encoding directionality is important for performance on the reported datasets.
- yNwD commends clarity around the transformation and the ternary incidence matrix idea.

## Outstanding / decisive concerns
- Baseline set is not appropriate for the paper’s stated goal (reaction classification).
  - yNwD’s main point: restricting comparisons to hypergraph baselines is unfair / incomplete for reaction classification; typical reaction prediction/classification includes strong chemoinformatics and modern neural baselines (ReactionFP; attention-based reaction mapping models; reaction-aware representation learning; transformer/set models).
  - ZxkP independently requests non-structural baselines (Deep Sets / Transformer encoder over fingerprint sets) to establish that the graph/hypergraph machinery is warranted. This is the key reason for rejection: without these comparisons, the claimed superiority is not well supported.
- Unclear positioning: method contribution vs. application contribution.
  - yNwD asks whether the novelty is primarily the ternary incidence matrix, the Laplacian theory, or the network design. The paper may be strong as a directed-hypergraph operator contribution, but it is framed and evaluated almost exclusively as a reaction classification paper—yet the evaluation does not meet the standards of that application domain.
- Generalization beyond reaction-type classification / beyond chemistry is not demonstrated.
  - ZxkP points out the absence of evidence on other directed-hypergraph tasks or non-chemical domains. If the intended contribution is a general directed-hypergraph spectral operator, the paper currently lacks breadth.
- Experimental rigor questions remain unanswered.
  - ZxkP asks about deduplication/leakage checks for the merged Dataset-2.
  - Sensitivity to architectural choices (depth/width/readouts) and fairness of baseline readouts are also open.
  - xuyY flags reproducibility issues (expired repo) and asks for deeper error analysis (confused classes).

xuyY is positive (8) but low confidence (2) and does not engage the baseline adequacy concern. yNwD is negative (2) with very high confidence (5), and ZxkP is borderline (4) with substantial confidence (4). Given the decisive nature of the baseline gap and the high-confidence reject, the safe meta-level decision is rejection.

**Reviewer Scores:**

Because there was no author response, I interpret this question as: if reviewers had fully discussed with each other, how might their scores shift based on shared concerns (especially baselines) and calibration.

`Reviewer yNwD (rating 2, confidence 5)`
- Expected change with full discussion: 2 → 2
- This reviewer’s concern is fundamental (missing standard baselines) and they express maximal confidence. Discussion would likely reinforce, not soften, their position.

·Reviewer xuyY (rating 8, confidence 2)·
- Expected change with full discussion: 8 → 4
- xuyY’s positive score appears driven by a high-level appreciation of the idea and proofs, but they explicitly have low confidence and did not scrutinize baseline fairness. In discussion, exposure to yNwD’s baseline critique and ZxkP’s request for strong non-graph baselines would likely lower their score toward borderline or below-threshold.

`Reviewer ZxkP (rating 4, confidence 4)`
- Expected change with full discussion: 4 → 2
- ZxkP is already slightly negative and highlights exactly the missing comparison that undermines the central claim. If discussion emphasizes that the paper is framed as reaction classification yet lacks standard baselines and broader validation, I would expect a downward shift.

---

### Decision · Program_Chairs · 2026-01-26

Reject